# GEFA: A General Feature Attribution Framework
# Using Proxy Gradient Estimation

**Yi Cai** [1]   **Thibaud Ardoin** [1]   **Gerhard Wunder** [1]

## Abstract

Feature attribution explains machine decisions by quantifying each feature's contribution. While numerous approaches rely on exact gradient measurements, recent work has adopted gradient estimation to derive explanatory information under query-level access, a restrictive yet more practical accessibility assumption known as the black-box setting. Following this direction, this paper introduces GEFA (Gradient-estimation-based Explanation For All), a general feature attribution framework leveraging proxy gradient estimation. Unlike the previous attempt that focused on explaining image classifiers, the proposed explainer derives feature attributions in a proxy space, making it generally applicable to arbitrary black-box models, regardless of input type. In addition to its close relationship with Integrated Gradients, our approach, a path method built upon estimated gradients, surprisingly produces unbiased estimates of Shapley Values. Compared to traditional sampling-based Shapley Value estimators, GEFA avoids potential information waste sourced from computing marginal contributions, thereby improving explanation quality, as demonstrated in quantitative evaluations across various settings.

## 1. Introduction

With the explosive growth of deep learning models, explainability has become an increasingly important research topic. While data-driven models excel in performance, their opaque nature, originating from the implicit learning processes, raises concerns and risks, particularly when deployed in critical domains such as medical diagnosis, finance, and autonomous driving. The demand for transparency has seen the development of various techniques, including feature attribution, which is the focus of this work.

Current attempts to determine feature attribution typically fall into two categories depending on the model accessibility assumption: the white-box and black-box methods. White-box approaches assume full access to a model, deriving explanations by investigating in detail the model's internal workings through, for example, analysis of gradients (Simonyan et al., 2014; Sundararajan et al., 2017) or supervision of information flow (Samek et al., 2021). Albeit beneficial to explanation procedures, the full accessibility assumption limits the applicability of white-box approaches under practical settings due to safety and security concerns. Models deployed for public use are usually wrapped by limited APIs and accessible only via queries. On the other hand, the black-box explainers, following the assumption of query-level access, determine feature attributions by analyzing the correlation between input features and model outcomes (Ribeiro et al., 2016). As a trade-off for the loosened accessibility assumption, black-box explanations tend to be less precise, especially when explaining models operating in high-dimension feature spaces. This is because inferring explanatory information indirectly from queries is computationally expensive, with the cost positively correlated to the dimensionality of the feature space.

Cai & Wunder (2024) sought to combine the strengths of both categories and proposed GEEX. Focusing on explaining image classifiers, GEEX delivers gradient-like explanations under a black-box setting based on estimated gradients, achieving a performance that matches white-box explainers. However, the method is limited to models operating on continuous input features and struggles with discrete or categorical features, such as text. This limitation arises from GEEX's reliance on path integrals, which are not well-defined in discrete feature spaces. Although applying GEEX at the embedding layer offers a plausible workaround, it arguably violates the black-box assumption by accessing internal model details during the transformation from the original feature space to an embedding space.

To bridge the gap in applicability to models operating on discrete data, this paper extends gradient-estimation-based

---

[1]Department of Mathematics and Computer Science, Freie Universität Berlin, Berlin, Germany. Correspondence to: Yi Cai <yi.cai@fu-berlin.de>.

*Proceedings of the 42$^{nd}$ International Conference on Machine Learning*, Vancouver, Canada. PMLR 267, 2025. Copyright 2025 by the author(s).

explanations and introduces GEFA[2] (Gradient-estimation-based Explanation For All), a general feature attribution framework built upon carefully designed proxy variables. These proxy variables facilitate the implementation of gradient estimation and path integral approximation, regardless of input formats. The proposed method comes with strong theoretical guarantees. First, GEFA is an unbiased calculator of Shapley Values (Shapley, 1953), demonstrated through rigorous mathematical proof. Compared to previous attempts in computing Shapley Values, GEFA reduces potential information waste in sampling-based estimations, which compute marginal contributions (Mitchell et al., 2022), and avoids calculations of factorials in the kernel method (Lundberg & Lee, 2017) for determining sample weights. Second, we show that our black-box explainer differs from Integrated Gradients (IG), a white-box approach by (Sundararajan et al., 2017), only in the path choice. It is proved that the two approaches become equivalent when their paths are aligned, emphasizing the connection between the gradient-estimation-based approach and actual gradients. Finally, we design a simple control variate that is mathematically proven to improve explanation quality under a simple and realistic assumption. Its effectiveness is validated through quantitative experiments across various settings.

## 2. Related Work

Gradients are widely used to allocate feature attributions in a white-box setting as they reveal a model's sensitivity to changes in feature values. In the early development of explainability, Simonyan et al. (2014) and Smilkov et al. (2017) interpreted gradients directly as explanations. Their methods retrieve explanatory information by tracing partial derivatives of a decision function with respect to its input features. Although adopting vanilla gradients is a reasonable starting point, gradients by themselves reflect local sensitivity and do not truthfully represent contributions of feature presence without a proper definition of feature absence.

IG (Sundararajan et al., 2017) addresses the limitation of vanilla gradients with a baseline point modeling feature absence. The approach integrates gradients over a straight-line path connecting the baseline and the explaining target, thereby capturing the overall impact of feature presence. Following work by Sturmfels et al. (2020) explored the impact of baseline choice and suggested adopting a distribution, rather than a deterministic instance, as the baseline (Erion et al., 2021). Other extensions of IG include decomposing noise directions from the path integral (Yang et al., 2023), refining explanations by filtering out high frequencies (Muzellec et al., 2024), and investigating feature interactions through the integration of second-order derivatives (Janizek et al., 2021). Parallel to these efforts in

improving the explanation procedure, Decker et al. (2024) demonstrated that a proper linear composition of explanations from various approaches yields provable improvements. The family of propagation-based methods (Montavon, 2019) represents a significant alternative white-box solution, which designs layer-wise back-propagation rules that explicitly utilize model architecture information for the retrieval of explanatory information. As this paper focuses primarily on gradient-based and gradient-like explanations, we refer interested readers to the survey by (Samek et al., 2021) for further details on relevance propagation.

Unlike white-box methods, which have direct access to model details, black-box explainers determine feature attributions by collecting and analyzing observations. The idea was proposed by LIME (Ribeiro et al., 2016), which generates queries by altering feature values of the original input and collects model responses to the perturbed instances. By solving a linear regression problem with the observed input-output pairs, LIME derives regressor coefficients as feature attributions. Subsequently, Lundberg & Lee (2017) proposed KernelSHAP, a kernel method that approximates Shapley Values using weighted linear regression. Additionally, Lundberg & Lee (2017) formalized the relationship between the feature attribution problem and cooperative game theory, strengthening the importance of Shapley Values in explainability.

Under the established framework of black-box approaches, succeeding works have aimed at improving query efficiency and explanation quality – long-standing challenges for black-box explainers. For example, Petsiuk et al. (2018) alleviated concerns about computational expenses by softly grouping input features via mask resizing. Dhurandhar et al. (2022) extended LIME with an adaptive neighborhood sampling scheme that constrains sampling to locally linear regions around the explicand. Similarly, Shrotri et al. (2022) and Dhurandhar et al. (2024) improved sampling efficiency by narrowing the search space. Parallel to the refinement of the sampling process, Frye et al. (2020) and Heskes et al. (2020) enhanced explanation quality by incorporating prior causal knowledge into the SHAP framework. Okhrati & Lipani (2021) leveraged the multilinear extension method from game theory literature (Owen, 1972) to develop a sampling-based explainer with reduced variance.

More recently, Cai & Wunder (2024) proposed GEEX, a black-box method imitating IG by integrating estimated gradients. While GEEX achieves white-box-level performance with only query-level access, its applicability is limited to continuous feature spaces due to the design of its search distribution. This paper extends gradient-estimation-based explanations into a general-purpose framework that is independent of specific input formats, broadening its applicability to a wider range of machine learning models.

---

[2]Code is available at: https://github.com/caiy0220/GEFA

# 3. Preliminary

## 3.1. Feature Attribution

Given a model function $f(\cdot)$, a target input (the explicand) $\boldsymbol{x} = (x_1, x_2, \ldots, x_p)$, and a predefined baseline $\mathring{\boldsymbol{x}} = (\mathring{x}_1, \mathring{x}_2, \ldots, \mathring{x}_p)$, an attribution method seeks a vector $\boldsymbol{\xi} \in \mathbb{R}^p$ that decomposes the total contribution to an inquired decision into feature attributions. Formally, this is represented as:

$$A_f : (\boldsymbol{x}, \mathring{\boldsymbol{x}}) \hookrightarrow (\xi_1, \xi_2, \ldots, \xi_p)$$

Throughout the paper, we mark vectors in bold and denote scalars with plain symbols.

As a result of allocating feature contributions, the attribution scores $\xi_i$ quantify the contribution of each feature $x_i$ to the model outcome $f(\boldsymbol{x})$. These scores should sum to the difference between the model outcome with all features present and the outcome with full feature absence, which is modeled by the baseline:

$$\sum_{i=0}^{p} \xi_i = f(\boldsymbol{x}) - f(\mathring{\boldsymbol{x}}) \tag{1}$$

Approaches complying with (1) are said to satisfy the property of *Completeness* – a fundamental axiom of feature attribution methods. Together with completeness, further properties are desired for feature attribution methods, which uphold their practical utility:

- *Sensitivity*: A feature should receive non-zero attribution if the difference between its value in the explicand and the baseline affects model outcomes.

- *Insensitivity*: The attribution should be zero for any feature on which the model is functionally independent.

- *Linearity*: The explanation for a linear composition of two functions should equal the weighted sum of the separate explanations for each function.

- *Symmetry*: If a function is symmetric in two variables $x_i$ and $x_j$, the attributions to the two features should be identical when the explicand-baseline pair satisfies $x_i = x_j$ and $\mathring{x}_i = \mathring{x}_j$.

## 3.2. Gradient Estimation under a Black-box Setting

In the context of feature attribution, a black box setting refers to query-level access, meaning that the model to be explained can only be accessed via its input and output interfaces. Indeed, lacking knowledge about the model's internal details prohibits the application of attribution methods that depend on exact measurements of gradients. However, gradients, which facilitate the derivation of feature attributions,

can still be estimated by evaluating model inputs and outputs. Defining a search distribution $\boldsymbol{\pi}(\cdot|\boldsymbol{x})$ parameterized by $\boldsymbol{x}$, the expected model outcome over $\boldsymbol{\pi}(\cdot|\boldsymbol{x})$ is given by:

$$J(\boldsymbol{x}) := \mathbb{E}_{\boldsymbol{\pi}(\boldsymbol{z}|\boldsymbol{x})}[f(\boldsymbol{z})] = \int f(\boldsymbol{z})\boldsymbol{\pi}(\boldsymbol{z}|\boldsymbol{x}) \, \mathrm{d}\boldsymbol{z} \tag{2}$$

where $\boldsymbol{z}$ indicates samples drawn from the search distribution. The gradient of the expected model outcome with respect to $\boldsymbol{x}$ is:

$$\nabla_{\boldsymbol{x}} J(\boldsymbol{x}) = \nabla_{\boldsymbol{x}} \int f(\boldsymbol{z})\boldsymbol{\pi}(\boldsymbol{z}|\boldsymbol{x}) \, \mathrm{d}\boldsymbol{z} \tag{3}$$

The above formula can be further simplified using the log-likelihood trick, under the assumption that both $f(\cdot)$ and $\boldsymbol{\pi}(\cdot|\boldsymbol{x})$ are continuously differentiable (Mohamed et al., 2020):

$$\begin{aligned} \nabla_{\boldsymbol{x}} J(\boldsymbol{x}) &= \int [f(\boldsymbol{z}) \cdot \nabla_{\boldsymbol{x}} \log \boldsymbol{\pi}(\boldsymbol{z}|\boldsymbol{x})]\boldsymbol{\pi}(\boldsymbol{z}|\boldsymbol{x}) \, \mathrm{d}\boldsymbol{z} \\ &= \mathbb{E}_{\boldsymbol{\pi}(\boldsymbol{z}|\boldsymbol{x})}[f(\boldsymbol{z}) \cdot \nabla_{\boldsymbol{x}} \log \boldsymbol{\pi}(\boldsymbol{z}|\boldsymbol{x})] \end{aligned} \tag{4}$$

The integral can be empirically approximated using a Monte Carlo estimator with a set of queries $\boldsymbol{Z} = \{\boldsymbol{z}|\boldsymbol{z} \sim \boldsymbol{\pi}(\cdot|\boldsymbol{x})\}$, leading to the typical score-function gradient estimator:

$$\boldsymbol{\eta}_{\boldsymbol{x}}(\boldsymbol{x}) := \nabla_{\boldsymbol{x}} J(\boldsymbol{x}) \approx \frac{1}{|\boldsymbol{Z}|} \sum_{\boldsymbol{z} \in \boldsymbol{Z}} f(\boldsymbol{z}) \cdot \nabla_{\boldsymbol{x}} \log \boldsymbol{\pi}(\boldsymbol{z}|\boldsymbol{x})$$

# 4. Gradient-Estimation-based Explanation

## 4.1. Gradient Estimation with Proxy Variables

Given the diverse nature of potential input features, sampling instances by perturbing feature values is not always straightforward. Instead of altering feature values by applying noises, we define the search distribution through a set of proxy variables $\boldsymbol{\alpha} = (\alpha_1, \alpha_2, \ldots, \alpha_p)$. The proxy vector $\boldsymbol{\alpha}$ shares the same size as the explicand, where each element $\alpha_i$ configures the presence probability of the corresponding explicand feature $x_i$. Feature presence and absence are modeled by the feature values of the explicand and the baseline, respectively. A point $\boldsymbol{x}(\boldsymbol{\alpha})$ in the continuous proxy space $\boldsymbol{\alpha} \in [0,1]^p$ describes a distribution, where each sample $\boldsymbol{z} \sim \boldsymbol{x}(\boldsymbol{\alpha})$ is given by:

$$z_i = \begin{cases} x_i & \text{if } \epsilon_i = 1 \\ \mathring{x}_i & \text{if } \epsilon_i = 0 \end{cases} \quad \forall i \in \{1, 2, \ldots, p\}$$

where $\boldsymbol{\epsilon} = (\epsilon_1, \epsilon_2, \ldots, \epsilon_p)$ denotes a binary mask sampled from a multivariate Bernoulli distribution parameterized by $\boldsymbol{\alpha}$, i.e. $\boldsymbol{\epsilon} \sim \text{Bernoulli}(\boldsymbol{\alpha})$. For ease of notation, we denote the feature selection process with a feature-wise combination operator $\oplus$. The operator indicates that a feature $z_i$ in

the sample $z$ takes the value of $x_i$ when the corresponding mask component $\epsilon_i = 1$, otherwise set to $\mathring{x}_i$:

$$z = \epsilon \circ x \oplus \bar{\epsilon} \circ \mathring{x}, \quad \epsilon \sim \text{Bernoulli}(\alpha)$$

The vector $\bar{\epsilon} = \mathbb{1}_p - \epsilon$ is the complement of $\epsilon$, and the operator $\circ$ indicates the element-wise product. A feature value is selected if the corresponding mask component equals one; otherwise, it remains undefined until assigned through the $\oplus$ operator. Please note that the feature selection operator is independent of feature types and is generally applicable as long as the explicand-baseline pair is specified. Given an explicand-baseline pair, the sampling of a query $z$ depends fully on the binary mask $\epsilon$, whose probability mass function is:

$$\pi(z|x(\alpha)) = \pi(\epsilon|\alpha) = \alpha^{\epsilon} \cdot (\mathbb{1}_p - \alpha)^{\bar{\epsilon}} \qquad (5)$$

Here, $\alpha^{\epsilon}$ is a shorthand for $(\alpha_1^{\epsilon_1}, \alpha_2^{\epsilon_2}, \dots, \alpha_p^{\epsilon_p})$. Substituting the distribution given by (5) into the search distribution $\pi$ in (4) yields an estimator for the gradient of $f(x(\alpha))$ w.r.t. the proxy variables $\alpha$:

$$\begin{aligned}
\eta_{\alpha}(x(\alpha)) &= \mathbb{E}_{\pi(z|x(\alpha))}[f(z) \cdot \nabla_{\alpha} \log \pi(z|x(\alpha))] \\
&= \mathbb{E}_{\pi(\epsilon|\alpha)}[f(\epsilon \circ x \oplus \bar{\epsilon} \circ \mathring{x}) \cdot \nabla_{\alpha} \log \pi(\epsilon|\alpha)] \\
&= \mathbb{E}_{\pi(\epsilon|\alpha)}[f(\epsilon \circ x \oplus \bar{\epsilon} \circ \mathring{x}) \cdot (\frac{\epsilon}{\alpha} - \frac{\bar{\epsilon}}{\mathbb{1}_p - \alpha})]
\end{aligned}$$
$$(6)$$

When referring to the logarithm of the probability vector $\pi$, the operation is applied element-wise to each vector component. Given that $\alpha$ represents the probabilities of feature presence, the output of $\eta_{\alpha}(x(\alpha))$ can be interpreted as the sensitivity of model outcomes to changes in feature presence.

### 4.2. Derivation of GEFA

In addition to promoting the derivation of the gradient estimator, the introduction of proxy variables facilitates the definition of path integrals for inputs with discrete features (e.g. text) when deriving feature attribution. Formally, let $\alpha(\cdot) = (\alpha_1, \dots, \alpha_p) : [0,1] \to [0,1]^p$ be a path in the proxy space, transitioning from the baseline $x(\alpha(0)) = x(\mathbb{0}_p) = \mathring{x}$ to the explicand $x(\alpha(1)) = x(\mathbb{1}_p) = x$, feature attributions are computed by integrating proxy gradient estimators along the path $\alpha(\gamma)$ for $\gamma \in [0,1]$. When taking the straightline path $\alpha(\gamma) = \gamma \cdot \mathbb{1}_p$, which is the only symmetry-preserving path (Sundararajan et al., 2017), the GEFA explainer is derived as follows:

$$\begin{aligned}
\xi &:= \int_0^1 \eta_{\alpha}(x(\gamma \cdot \mathbb{1}_p)) \, d\gamma \\
&= \int_0^1 \mathbb{E}_{\pi(\epsilon|\gamma \cdot \mathbb{1}_p)}[f(\epsilon \circ x \oplus \bar{\epsilon} \circ \mathring{x}) \cdot (\frac{\epsilon}{\gamma} - \frac{\bar{\epsilon}}{1 - \gamma})] \, d\gamma
\end{aligned}$$
$$(7)$$

In practice, (7) can be approximated with a Monte-Carlo estimator, given a budget of $n$ queries:

$$\xi \approx \frac{1}{n} \sum_{\gamma \sim \mathcal{U}_{[0,1]}} \sum_{\pi(\epsilon|\gamma \cdot \mathbb{1})} f(\epsilon \circ x \oplus \bar{\epsilon} \circ \mathring{x}) \cdot (\frac{\epsilon}{\gamma} - \frac{\bar{\epsilon}}{1-\gamma}) \ (8)$$

**Theorem 1.** *GEFA satisfies the properties of Completeness, Sensitivity, Insensitivity, Linearity, and Symmetry.*

Appendix A.2 elaborates on these properties and the corresponding proofs, derived from the gradient estimation perspective following (7). Beyond the proven properties, we surprisingly find that GEFA, an approach derived from a proxy gradient estimator, offers an alternative for computing Shapley Values, as stated in Theorem 2.

**Theorem 2.** *Feature attributions determined by GEFA are exactly Shapley Values.*

The claim in Theorem 2 is mathematically rigorously proved, please refer to Appendix A.1 for further details. Being an unbiased calculator of Shapley Values also explains the many properties held by GEFA.

While also producing an unbiased approximation of Shapley Values, GEFA differs from other sampling-based attempts by simplifying the sampling process. Specifically, the computation of (8) does not rely on marginal contributions, thus avoiding potential information waste during approximation. Let $z_S$ denote a query with $S$ being the set of indices corresponding to the present features. In GEFA, each query $z_S$ contributes to the attribution estimates of every feature $x_i$ for $i \in \{1, 2, \dots, p\}$, regardless of the existence of paired samples $z_{S \cup \{i\}}$ (when $i \notin S$) or $z_{S \setminus \{i\}}$ (when $i \in S$) that are typically required for computing marginal contributions. Algorithm 1 summarizes the overall explanation scheme derived from (8).

---

**Algorithm 1** GEFA Explanation Scheme

---

**Input:** $x$: the explicand; $\mathring{x}$: the baseline;
**Output:** $\xi$: feature attribution scores;
  1: $\xi = \mathbb{0}_p$               # *Estimator initialization*
  2: **while** Query budget available **do**
  3:     $\gamma \sim \mathcal{U}_{[0,1]}$       # *Proxy path point sampling*
  4:     $\epsilon \sim \pi(\cdot|\gamma \cdot \mathbb{1}_p)$         # *Mask sampling*
  5:     $z = \epsilon \circ x \oplus \bar{\epsilon} \circ \mathring{x}$     # *Query construction*
  6:     $\xi \mathrel{+}= \frac{1}{n} \cdot f(z) \cdot (\frac{\epsilon}{\gamma} - \frac{\bar{\epsilon}}{1-\gamma})$ # *Observation collection*
  7: **end while**
  8: **return** $\xi$

---

### 4.3. Variance Reduction

Deriving the explainer from a score-function gradient estimator allows the application of variance reduction techniques in the gradient estimation literature. Specifically,

we construct a control variate to reduce estimation variance under the assumption that target model outcomes are correlated with the number of present features, denoted by $\|\boldsymbol{\epsilon}\|_1 = \sum_{i=1}^{p} \epsilon_i$. Assumption 3 formally states the condition required for the *validity* of the designed control variate.

**Assumption 3.** For any explicand-baseline pair that satisfies $f(\boldsymbol{x}) \neq f(\mathring{\boldsymbol{x}})$, the correlation between the number of present features and the corresponding model outcomes must be non-zero.

In practice, we argue that the above assumption generally holds for any properly trained model that bases its predictions on evidence observed in its inputs. A higher ratio of presented features increases the likelihood of including prediction-relevant components, which steers the model's predictions from $f(\mathring{\boldsymbol{x}})$ toward $f(\boldsymbol{x})$. Therefore, a non-zero correlation is to be expected. Based on Assumption 3, the control variate is constructed as a function of $\|\boldsymbol{\epsilon}\|_1$:

$$h(\|\boldsymbol{\epsilon}\|_1) = \begin{cases} 0 & \text{if } \|\boldsymbol{\epsilon}\|_1 = 0, \; p \\ \|\boldsymbol{\epsilon}\|_1/p - 0.5 & \text{else} \end{cases} \quad (9)$$

Adding the control variate, weighted by a coefficient $\beta$, to the target function gives:

$$\tilde{f}(\boldsymbol{\epsilon} \circ \boldsymbol{x} \oplus \bar{\boldsymbol{\epsilon}} \circ \mathring{\boldsymbol{x}}) = f(\boldsymbol{\epsilon} \circ \boldsymbol{x} \oplus \bar{\boldsymbol{\epsilon}} \circ \mathring{\boldsymbol{x}}) - \beta \cdot h(\|\boldsymbol{\epsilon}\|_1) \quad (10)$$

Replacing $f(\cdot)$ in (7) accordingly with the updated $\tilde{f}(\cdot)$ yields the variant G$\tilde{\text{E}}$FA:

$$\tilde{\boldsymbol{\xi}} = \int_0^1 \mathbb{E}_{\boldsymbol{\pi}(\boldsymbol{\epsilon}|\gamma \cdot \mathbb{1}_p)}[\tilde{f}(\boldsymbol{\epsilon} \circ \boldsymbol{x} \oplus \bar{\boldsymbol{\epsilon}} \circ \mathring{\boldsymbol{x}}) \cdot (\frac{\boldsymbol{\epsilon}}{\gamma} - \frac{\bar{\boldsymbol{\epsilon}}}{1 - \gamma})] \, \mathrm{d}\gamma \quad (11)$$

**Theorem 4.** *The unbiasedness of $\tilde{\boldsymbol{\xi}}$ remains intact after the introduction of the control variate $h(\cdot)$.*

Appendix A.3 provides the proof of Theorem 4, along with the derivation and further details of $h(\cdot)$. The variance reduction effect is optimized when the weighting coefficient is set to $\beta^* = \text{Cov}(f, h)/\text{Var}(h)$, as shown by Appendix A.3.2. To compute $\beta^*$, the variance of the control variate can be derived in closed form, and the covariance, though not explicitly given, can be empirically estimated with existing observations used for attribution computation (Mohamed et al., 2020). The optimal weighting coefficient also ensures that the control variate has no negative effect even in the worst case where Assumption 3 is violated, as the added term $h(\cdot)$ will be eliminated by the zeroness of $\beta^*$, which results from the zero correlation $\text{Cov}(f, h) = 0$.

### 4.4. Relation to Integrated Gradients

Since the proposed method is built upon estimated gradients, this section further explores its relationship to IG[3], which

---

[3]By considering IG, we omit the practical difficulty that discrete features are usually not differentiable in their original forms, thus requiring additional pre-/post-processing steps.

utilizes actual gradients. The equivalence between GEFA and IG does not hold when both take a straightline path, as GEFA's path is constructed in the proxy space, which differs from the original feature space. However, their relationship becomes clearer when both explainers follow a monotonic path along the edges of their respective spaces. Along an edge path, integration moves step-by-step from one vertex $\boldsymbol{z}_{\boldsymbol{S}}$ in the feature/proxy space to an adjacent vertex $\boldsymbol{z}_{\boldsymbol{S} \cup \{i\}}$ that differs in only one feature.

**Theorem 5.** *GEFA and IG are equivalent when taking the same edge path. Averaging their results over all $p!$ unique edge paths converges to the outcome of GEFA following the straightline path in the proxy space.*

It can be easily shown that, when following the same permutation order, GEFA and IG both compute the marginal contribution of a feature $x_i$, namely $f(\boldsymbol{z}_{\boldsymbol{S}}) - f(\boldsymbol{z}_{\boldsymbol{S} \cup \{i\}})$, conditioned on a set of present features $\{x_j | j \in \boldsymbol{S}\}$. Given the fact that GEFA is an unbiased estimator of Shapley Values, concluding Theorem 5 is not surprising – averaging marginal contributions is the typical solution for computing Shapley Values. Please refer to Appendix A.4 for the detailed proof. The close relationship between IG and Shapley Values is consistent with previous claims by Sundararajan & Najmi (2020). Furthermore, Theorem 5 motivates the choice of the straightline path along the diagonal of the proxy space, converting the problem of averaging estimates over numerous edge paths to estimating attributions on one specific path.

## 5. Experiments

### 5.1. Experimental Setting

To show GEFA's applicability and effectiveness across various scenarios, we evaluate its performance on representative tasks involving discrete and continuous features: text and image classification.

*Dataset*: Three datasets are adopted for text classification tasks: *Amazon Review Polarity* (McAuley & Leskovec, 2013), *STS-2*, and *QNLI* (Wang et al., 2019). The image classification task is set up with *ImageNet* (Russakovsky et al., 2015), whose high-dimensional input feature space poses challenges to black-box explainers that derive feature attributions through queries.

*Classifier*: We fine-tune a pretrained version of *BERT* on the Amazon dataset and use *Llama3.2-3B-Instruct* as a zero-shot learner for STS-2 and QNLI. Llama3.2 is configured as a classifier through prompt engineering, enabling the focus on the standard feature attribution setting for classification tasks. For ImageNet, the pretrained *InceptionV3* and *Vision Transformer* (*ViT*) models are adopted without further fine-tuning. The selected models represent a variety of ar-

chitectures and scales, demonstrating that the explanation quality of the proposed method is independent of specific model choices. Additional details about the tested models, including the prompts used, can be found in Appendix B.

*Evaluation via manipulation*: Despite explainability being a widely studied topic, there is still no consensus for quantitative evaluations of explanation quality due to the lack of ground truth explanations. Adapting to the practical difficulty, evaluation via deletion (Samek et al., 2016) quantifies an explainer's performance indirectly by assessing the effectiveness of feature removal guided by explanations. Following the intuition that removing relevant features should induce significant changes in prediction results, the evaluation scheme repeatedly removes features in descending order according to their attribution scores. The area over the perturbation curve (AOPC), drawn by the sequence of prediction outcomes after feature removal, quantifies explanation quality. A larger area indicates a more informative explanation that amplifies the impact of the deletion process. Formally, let $x^{(k)}$ denote a manipulated version of the explicand with $k$ features removed, the normalized AOPC is computed by:

$$\text{nAOPC} = \frac{1}{p} \sum_{k=1}^{p} (1 - \frac{f(x^{(k)})}{f(x)})$$

When choosing the deletion-based evaluation scheme, we acknowledged concerns about its validity, since the recursive deletion process may shift the manipulated explicand away from the target data manifold, potentially introducing an additional source of prediction change (Hooker et al., 2019). To address this issue, Appendix C provides a detailed discussion on the validity of the adopted evaluation scheme, supported by empirical results demonstrating its alignment with the alternative retraining scheme (Hooker et al., 2019) – a computationally expensive evaluation approach designed to mitigate the out-of-distribution concern.

*Competitors*: We consider several feature attribution methods closely related to the proposed method. The competitors include two gradient-based approaches assuming white-box access: **VG** (Vanilla Gradient) and **IG** (Integrated Gradient); and three black-box explainers: **KSHAP** (KernelSHAP), **PSHAP** (PartitionSHAP), and **GEEX** (Gradient-Estimation-based Explanation). Appendix B.3 details these competitors and their implementations. The selected competitors are evaluated following the aforementioned evaluation scheme and compared to the two variants of the proposed methods: GEFA and GẼFA, representing the versions without and with the control variate, respectively. In addition to the listed explainers, random feature removal (abbreviated as **Random**) is included as a baseline competitor. It removes features randomly, simulating the absence of explanatory information. Any explainer delivering valid explanations should achieve a higher nAOPC score than random removal.

## 5.2. Explaining Text Classifiers

When applying feature attribution methods to text classifiers, black-box approaches like GEFA offer greater flexibility in representing feature absence, as they construct synthetic instances in the original text space for querying. Unlike models for other tasks, text classifiers typically accept variable-length inputs, simplifying the definition of absence. Therefore, we adopt the *empty token* as the baseline for the black-box explainers. As such, feature absence is explicitly represented by removing the corresponding feature, rather than replacing its value with some manually defined baseline value. In contrast, white-box approaches relying on back-propagation stick to the compromised definition of feature absence – replacing features with default values. This limitation arises because exact gradient measurement via back-propagation requires placeholders in the input for the propagation process. Since text sequences are non-differentiable, the baseline value must be defined in the embedding space. Lacking precise knowledge about the embedding manifold, we empirically choose the zero embedding vector to represent feature absence for VG and IG. Embedding-level attributions are aggregated for each token to produce the final explanations.

Table 1 presents the nAOPC scores of the competitors tested across various text classification settings. Each row in the table corresponds to a test case specified by the dataset and the classifier. For all test cases, the query budget for the black-box explainers is $500$, given the relatively smaller feature space; the interpolation step for IG is set to $50$. Please note that GEEX is excluded from this part of the evaluation due to its incompatibility with models operating on discrete feature space, as previously discussed in Section 1.

Notably, the explanations by VG barely deliver any valid information as evidenced by its performance, which matches the level of random removal across all three tested settings. This observation suggests that directly interpreting gradients as explanations is inappropriate, since the raw gradient itself merely reveals a model's local sensitivity to a feature, which does not necessarily associate with the feature's contribution to a prediction. The qualitative example in Figure 1 showcases the failure of VG to identify relevant features, in contrast to IG and GẼFA. While disagreements in attributions exist between IG and GẼFA, their explanations agree on the primary evidence supporting a positive prediction. VG produces a contradictory result by identifying 'pain' as a positively contributing feature and assigning importance to a stop word 'that' as evidence in sentiment analysis, which appears less reasonable. Among the group of black-box explainers, GẼFA achieves the best performance over other sampling-based Shapley Value estimators. We attribute the improvement to the mitigation of information waste during estimation and the variance reduction through the designed

Table 1: The nAOPCs reported on text classification tasks, higher is better.

| Dataset | Model | VG | IG | KSHAP | PSHAP | GEFA | $\widetilde{\text{GEFA}}$ | Random |
|---------|-------|-----|-----|-------|-------|------|------|--------|
| Amazon | BERT | 0.1823 | 0.6677 | 0.6014 | 0.6592 | 0.7120 | **0.7366** | 0.1908 |
| SST-2 | Llama3.2-3B | 0.2518 | 0.3664 | 0.5386 | 0.5122 | 0.5460 | **0.5706** | 0.2472 |
| QNLI | Llama3.2-3B | 0.2411 | 0.2985 | 0.4106 | 0.4280 | 0.4472 | **0.4740** | 0.2271 |

*The overall best performances are in **bold** and the highest scores among black-box explainers are underlined.

ive got a lamp in the corner of my room behind my desk thats a complete pain in the arse to turn on and off. ive been using this with the lamp for a month now and it works perfectly. added a little velcro and now i have a light switch where ever i want. under my desk, shelf, etc.

(a) VG

ive got a lamp in the corner of my room behind my desk thats a complete pain in the arse to turn on and off. ive been using this with the lamp for a month now and it works perfectly. added a little velcro and now i have a light switch where ever i want. under my desk, shelf, etc.

(b) IG

ive got a lamp in the corner of my room behind my desk thats a complete pain in the arse to turn on and off. ive been using this with the lamp for a month now and it works perfectly. added a little velcro and now i have a light switch where ever i want. under my desk, shelf, etc.

(c) GEFA

Figure 1: Feature attributions for BERT derived from three selected explainers. The results are visualized by attribution maps, where blue and red background colors indicate contributions to positive and negative sentiments, respectively, with color intensity reflecting the magnitude of the attribution scores.

control variate. The comparison between the two GEFA variants further highlights the effectiveness of the control variate, whose design follows a simple intuition.

Furthermore, despite guaranteeing limited access, $\widetilde{\text{GEFA}}$ outperforms the gradient-based explainers across all settings, with a pronounced edge in the two complicated test cases established on Llama3.2. While the higher figures indeed emphasize the effectiveness of the proposed method, these results do *not* simply suggest that $\widetilde{\text{GEFA}}$ is the better option for explaining text models, particularly given the overall advantage of the black-box approaches over the white-box explainers. Instead, we interpret the performance difference as a consequence of the distinct *absence representations*. For the gradient-based approaches, selecting a proper baseline is particularly challenging in the embedding space, which is high-dimensional and often incomprehensible to humans. These methods are constrained to adopt the compromised absence definition in the embedding space due to their inherent explanation mechanisms, whereas the query-based approaches can conveniently take the natural absence definition as discussed before. Although not contradicting the effectiveness of white-box solutions, the practical difficulty of baseline selection reflected by the observations should be considered a non-trivial factor when deciding tools for text model explainability.

### 5.3. Explaining Image Classifiers

We perform the same evaluation with image classifiers to assess the quality of explanations derived from proxy gradient estimators in continuous feature spaces. The query budget of the black-box approaches is increased to 5000 due to the considerably larger input feature spaces, which are $299 \times 299$ and $224 \times 224$ for InceptionV3 and ViT,

respectively. KSHAP is excluded from this evaluation because solving the linear regression requires a query budget at least matching the dimensionality of the input feature space, which is impractical for models with high-dimensional inputs. Since image classifiers cannot process incomplete inputs, feature absence in this context is represented by replacing features with a baseline value. In accordance with the suggestion by Sturmfels et al. (2020), we use a blurred version of the explicand as the baseline.

As shown in Table 2, the performance and relative ranking of the competitors are consistent with the observations and analysis from the previous experiment. With the feature absence definition aligned, IG guides the most effective deletion process. Closely following the best-performing white-box approach, $\widetilde{\text{GEFA}}$ retains competitive performance in the high-dimensional settings. It is noteworthy that, when explaining the image classifiers, the control variate yields larger performance improvements compared to the text settings. This is attributed to the stronger covariances between the control variate and the decision function (see Appendix D.3 for more details). In image classification, each feature (a pixel) contributes minimally to the overall prediction and typically carries less semantic weight, in contrast to features in sentiment analysis. Contextual dependencies on specific tokens (e.g. negation or irony) in sentiment analysis partially undermine the validity of Assumption 3. With the variance of the control variate remaining constant, the increased amplitude of the covariance between $f(\cdot)$ and $h(\cdot)$ contributes positively to the variance reduction effect, as detailed in Appendix A.3, thus enhancing the overall quality of explanations.

Additionally, the comparison between the proposed approach and GEEX, the other gradient-estimation-based

Table 2: The nAOPCs reported on InceptionV3 and ViT for ImageNet, higher is better.

| Dataset | Model | VG | IG | PSHAP | GEEX | GEFA | GẼFA | Random |
|---------|-------|-----|-----|-------|------|------|------|--------|
| ImageNet | InceptionV3 | 0.4570 | **0.8805** | 0.7753 | 0.7952 | 0.8352 | 0.8747 | 0.4003 |
| ImageNet | ViT | 0.3280 | **0.8571** | 0.6579 | 0.7539 | 0.7638 | 0.8096 | 0.3310 |

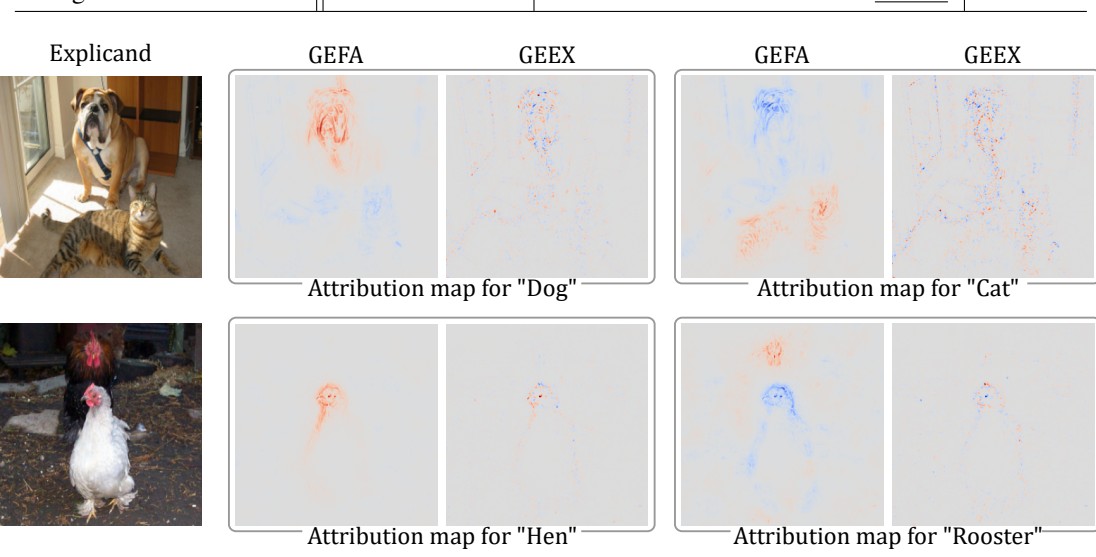

Figure 2: Feature attributions for InceptionV3 illustrating the evidence supporting predictions for specific classes. Red pixels indicate support for the prediction of the target class, whereas blue pixels oppose it. Color intensity indicates the magnitude of attributions.

method, is worth mentioning. Binarized feature value sampling by GEFA induces more significant prediction changes than the small Gaussian noise perturbations used by GEEX, facilitating more effective gradient estimation. In the experiments, we find that explanations by GEEX are more sensitive to low-level features that are generally informative, such as object contours, but they struggle to differentiate which specific class those features contribute to. Examples listed in Figure 2 demonstrate that GEFA distinguishes features relevant to specific classes, whereas GEEX fails to do so. In the "dog-cat" example, although there are differences in GEEX's explanations between the selected classes, pixels relevant to "dog" are consistently highlighted, making it difficult to comprehend their relationship to specific classes. On the contrary, the explanations by GEFA clearly differentiate the contributions of the same features in different contexts, as indicated by the pixel coloring. Pixels representing "dog" and "cat" exhibit conflicting contributions, reflecting the effect of the softmax layer concatenated before the final output layer – the probability increase of one class undermines the other. Similar observations can be obtained in the "rooster-hen" example, where GEEX concentrates on one object and overlooks the fact that the model can distinguish between a rooster and a hen, as demonstrated by GEFA.

## 6. Conclusion

In this paper, we propose GEFA, a model-agnostic feature attribution framework based on proxy gradient estimation. By structuring the explanation process in the proxy space, GEFA is generally applicable for explaining arbitrary classifiers, regardless of their input feature types. Backed by rigorous theoretical analysis, the proposed method significantly improves the quality of black-box explanations and, in certain circumstances, even surpasses white-box approaches with a limited query budget. As a general framework, GEFA holds significant potential for integration with existing techniques to further enhance sampling efficiency (Shrotri et al., 2022; Dhurandhar et al., 2024) and explanation quality (Frye et al., 2020; Heskes et al., 2020).

In addition to the primary focus on the typical feature attribution problem, the experiment section explores and demonstrates the effectiveness of the proposed method in explaining LLMs for simplified test cases, namely deriving feature attributions for the next predicted token. A promising future direction is to extend the described framework to explain the general generative behavior of LLMs. Operating in an auto-regressive manner, LLMs pose unique challenges to existing attribution methods due to the delayed observation of outcomes – gradients of the entire output sequence cannot

be back-propagated. However, analogous to the application of gradient estimation in reinforcement learning, GEFA, built upon proxy gradient estimation, has the potential to deal with delayed observations, thereby analyzing complex text sequences in their entirety instead of focusing on individual output tokens. With future adjustments to summarize observed text sequences, GEFA could deliver more holistic explanations for LLMs' outcomes.

## Acknowledgements

Yi Cai, Thibaud Ardoin, and Gerhard Wunder were supported by the Federal Ministry of Education and Research of Germany (BMBF) in the program of "Souver¨an. Digital. Vernetzt.", joint project "AIgenCY: Chances and Risks of Generative AI in Cybersecurity", project identification number 16KIS2013. Gerhard Wunder was also supported by BMBF joint project "6G-RIC: 6G Research and Innovation Cluster", project identification number 16KISK020K.

## Impact Statement

This paper presents a general-purpose feature attribution framework for explaining machine decisions. By leveraging proxy gradient estimation, the proposed method extends its applicability to a wide range of models under the practical black-box accessibility assumption, making it suitable for diverse real-world applications. We believe that this framework holds the potential to enhance the supervision and verification of machine decisions, thereby contributing to more transparent and trustworthy human-machine interactions.

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

## A. Mathematical Proofs

### A.1. Proof of GEFA's Equivalence to Shapley Values

We start with proving Theorem 2, as the notations introduced during the proof facilitate the proof of the properties listed in Theorem 1. To show that the attributions delivered by GEFA are exact Shapley Values, the goal is to demonstrate the following equivalence:

$$\xi_i = \sum_{\mathbf{S} \subseteq \{1,2,\dots,p\} \setminus \{i\}} \frac{|\mathbf{S}|!(p - |\mathbf{S}| - 1)!}{p!} \cdot (f(\mathbf{z}_{\mathbf{S} \cup \{i\}}) - f(\mathbf{z}_{\mathbf{S}})) = Sh_i$$

where $\mathbf{z}_{\mathbf{S}}$ denotes a query with $\mathbf{S}$ being the set of indices corresponding to the present features.

*Proof of Theorem 2.* Let $\mathbf{z}_{\mathbf{S}}$ be a query, the probability of sampling $\mathbf{z}_{\mathbf{S}}$ over the integration path is:

$$p(\mathbf{z}_{\mathbf{S}} | \mathbf{x}) = \int_0^1 \gamma^{|\mathbf{S}|} \cdot (1 - \gamma)^{(p - |\mathbf{S}|)} \, d\gamma$$

For a feature $x_i$, where $i \notin \mathbf{S}$, the contribution of the query to the computation of the corresponding attribution, noted as $w_i^{\mathbf{z}_{\mathbf{S}}}$, is:

$$
\begin{aligned}
w_i^{\mathbf{z}_{\mathbf{S}}} &= \int_0^1 \gamma^{|\mathbf{S}|} \cdot (1 - \gamma)^{(p - |\mathbf{S}|)} \cdot f(\mathbf{z}_{\mathbf{S}}) \cdot \left(\frac{0}{\gamma} + \frac{1 - 0}{1 - \gamma}\right) d\gamma \\
&= -\int_0^1 \gamma^{|\mathbf{S}|} \cdot (1 - \gamma)^{(p - |\mathbf{S}| - 1)} \cdot f(\mathbf{z}_{\mathbf{S}}) \, d\gamma \\
&= -\frac{|\mathbf{S}|!(p - |\mathbf{S}| - 1)!}{p!} \cdot f(\mathbf{z}_{\mathbf{S}}) \quad \text{(Beta-function)}
\end{aligned}
$$

Similarly, the weight of the query $\mathbf{z}_{\mathbf{S} \cup \{i\}}$ that differs from $\mathbf{z}_{\mathbf{S}}$ only in the $i$-th feature is:

$$
\begin{aligned}
w_i^{\mathbf{z}_{\mathbf{S} \cup \{i\}}} &= \int_0^1 \gamma^{|\mathbf{S}| + 1} \cdot (1 - \gamma)^{(p - |\mathbf{S}| - 1)} \cdot f(\mathbf{z}_{\mathbf{S} \cup \{i\}}) \cdot \left(\frac{1}{\gamma} + \frac{1 - 1}{1 - \gamma}\right) d\gamma \\
&= \frac{|\mathbf{S}|!(p - |\mathbf{S}| - 1)!}{p!} \cdot f(\mathbf{z}_{\mathbf{S} \cup \{i\}})
\end{aligned}
$$

Summing over all possible combinations of feature presences (excluding $x_i$), yields $\xi_i$:

$$
\begin{aligned}
\xi_i &= \sum_{\mathbf{S} \subseteq \{1,2,\dots,p\} \setminus \{i\}} w_i^{\mathbf{z}_{\mathbf{S}}} + w_i^{\mathbf{z}_{\mathbf{S} \cup \{i\}}} \\
&= \sum_{\mathbf{S} \subseteq \{1,2,\dots,p\} \setminus \{i\}} \frac{|\mathbf{S}|!(p - |\mathbf{S}| - 1)!}{p!} \cdot (f(\mathbf{z}_{\mathbf{S} \cup \{i\}}) - f(\mathbf{z}_{\mathbf{S}})) \\
&\Leftrightarrow Sh_i
\end{aligned}
$$

$\square$

### A.2. Proofs of Claimed Properties

It is not surprising that GEFA aligns with the properties held by Shapley Values as an unbiased calculator. This section details the proof of these properties from the gradient estimator perspective as an alternative to the derivation from the typical computation of Shapley Values in the form of marginal contributions.

#### A.2.1. COMPLETENESS AND SENSITIVITY

**Completeness** requires the equivalence between the sum of allocated feature attributions and the difference in prediction results made by full feature presence as stated in (1).

*Proof of Completeness.* The contribution of a sample $\boldsymbol{z_S}$ to attribution estimation in GEFA can be divided into two parts, the contribution with a positive sign $w_{i \in \boldsymbol{S}}$ to the present features $\{x_i | i \in S\}$, and the contribution with a negative sign $w_{i \notin \boldsymbol{S}}$ to the absent features. According to (8), the contribution is computed by:

$$w_{i \in \boldsymbol{S}} := f(\boldsymbol{z_S}) \cdot \frac{1}{\gamma}$$

$$w_{i \notin \boldsymbol{S}} := -f(\boldsymbol{z_S}) \cdot \frac{1}{1 - \gamma}$$

Considering the likelihood of $\boldsymbol{z_S}$ being sampled, the total positive contribution $w_{\boldsymbol{S}}^{\oplus}$ can be computed by:

$$
\begin{aligned}
w_{\boldsymbol{S}}^{\oplus} &= \int_0^1 \gamma^{|\boldsymbol{S}|} \cdot (1 - \gamma)^{(p - |\boldsymbol{S}|)} \cdot \left(\sum_{i \in \boldsymbol{S}} w_{i \in \boldsymbol{S}}\right) \mathrm{d}\gamma \\
&= \int_0^1 \gamma^{|\boldsymbol{S}|} \cdot (1 - \gamma)^{(p - |\boldsymbol{S}|)} \cdot f(\boldsymbol{z_S}) \cdot \frac{|S|}{\gamma} \, \mathrm{d}\gamma \\
&= \frac{(|S| - 1)!(p - |S|)!}{p!} \cdot f(\boldsymbol{z_S}) \cdot |S| \qquad \text{(Beta-function)} \\
&= \frac{|S|!(p - |S|)!}{p!} \cdot f(\boldsymbol{z_S})
\end{aligned}
$$

Similarly, the total negative contribution is:

$$
\begin{aligned}
w_{\boldsymbol{S}}^{\ominus} &= -\int_0^1 \gamma^{|\boldsymbol{S}|} \cdot (1 - \gamma)^{(p - |\boldsymbol{S}|)} \cdot f(\boldsymbol{z_S}) \cdot \frac{p - |S|}{1 - \gamma} \, \mathrm{d}\gamma \\
&= -\frac{(|S|)!(p - |S| - 1)!}{p!} \cdot f(\boldsymbol{z_S}) \cdot (p - |S|) \\
&= -\frac{|S|!(p - |S|)!}{p!} \cdot f(\boldsymbol{z_S})
\end{aligned}
$$

The two parts of contributions cancel out as $w_{\boldsymbol{S}}^{\oplus} + w_{\boldsymbol{S}}^{\ominus} = 0$, with the only two exceptions when $\boldsymbol{S} = \emptyset$ or $\boldsymbol{S} = \{1, 2, \ldots, p\}$, whose contribution only has the negative/positive part:

$$w_{\emptyset}^{\oplus} + w_{\emptyset}^{\ominus} = 0 - f(\mathring{\boldsymbol{x}})$$
$$w_{\{1,2,\ldots,p\}}^{\oplus} + w_{\{1,2,\ldots,p\}}^{\oplus} = f(\boldsymbol{x}) - 0$$

Computing the sum of feature attributions by summarizing sample contributions results in:

$$\sum_{i=1}^p \xi = \sum_{\boldsymbol{S} \subseteq \{1,2,\ldots,p\}} (w_{\boldsymbol{S}}^{\oplus} + w_{\boldsymbol{S}}^{\ominus}) = f(\boldsymbol{x}) - f(\mathring{\boldsymbol{x}})$$

$\square$

*Sensitivity* is guaranteed by the satisfaction of completeness.

### A.2.2. INSENSITIVITY

**Insensitivity** is also known as *Dummy*, which requires the attribution score to be zero for any feature on which the target model is not functionally dependent. Definition 6 formally describes functional independence.

**Definition 6.** A function is said to be *functionally independent* of a feature if the prediction results are always the same for any sample pair that differs only in that feature.

*Proof of Insensitivity.* Let $x_i$ be the dummy feature, the proxy gradient estimator of that feature on the straightline path is:

$$\eta_{\alpha_i}(\boldsymbol{x}(\gamma \cdot \mathbb{1}_p)) = \mathbb{E}_{\boldsymbol{\pi}(\boldsymbol{\epsilon}|\gamma\cdot\mathbb{1}_p)}[f(\boldsymbol{\epsilon}\circ\boldsymbol{x}\oplus\bar{\boldsymbol{\epsilon}}\circ\mathring{\boldsymbol{x}})\cdot(\frac{\epsilon_i}{\gamma}-\frac{\bar{\epsilon}_i}{1-\gamma})]$$

Using $\boldsymbol{\pi}(\boldsymbol{\epsilon}_{\backslash i}|\gamma\cdot\mathbb{1}_{p-1})$ as a shorthand for the feature value sampling process excluding the $i$-th feature, the expectation can be expanded to the following form due to the independent sampling processes of different features:

$$\eta_{\alpha_i}(\boldsymbol{x}(\boldsymbol{\alpha})) = \mathbb{E}_{\boldsymbol{\pi}(\boldsymbol{\epsilon}_{\backslash i}|\gamma\cdot\mathbb{1}_{p-1})}\Big[\mathbb{E}_{\boldsymbol{\pi}(\epsilon_i|\gamma)}[f(\boldsymbol{\epsilon}\circ\boldsymbol{x}\oplus\bar{\boldsymbol{\epsilon}}\circ\mathring{\boldsymbol{x}})\cdot(\frac{\epsilon_i}{\gamma}-\frac{\bar{\epsilon}_i}{1-\gamma})]\Big]$$

The condition of functional independence of $x_i$ yields:

$$\eta_{\alpha_i}(\boldsymbol{x}(\boldsymbol{\alpha})) = \mathbb{E}_{\boldsymbol{\pi}(\boldsymbol{\epsilon}_{\backslash i}|\gamma\cdot\mathbb{1}_{p-1})}\Big[\mathbb{E}_{\boldsymbol{\pi}(\epsilon_i|\gamma)}[f(\boldsymbol{\epsilon}\circ\boldsymbol{x}\oplus\bar{\boldsymbol{\epsilon}}\circ\mathring{\boldsymbol{x}})]\cdot\underbrace{\mathbb{E}_{\boldsymbol{\pi}(\epsilon_i|\gamma)}[(\frac{\epsilon_i}{\gamma}-\frac{\bar{\epsilon}_i}{1-\gamma})]}_{=0}\Big]$$

$$= 0$$

The explainer integrating over $\eta_{\alpha_i}(\boldsymbol{x}(\boldsymbol{\alpha}))$ also produces zero, namely $\xi_i = 0$. □

### A.2.3. LINEARITY

For any two functions $f_1(\cdot)$ and $f_2(\cdot)$, **Linearity** requires the explanation for the linear composition of the two functions equaling the weighted sum of the separate explanations for them:

$$\boldsymbol{\xi}^{(af_1+bf_2)} = a\cdot\boldsymbol{\xi}^{(f_1)} + b\cdot\boldsymbol{\xi}^{(f_2)}$$

*Proof of Linearity.*

$$\boldsymbol{\xi}^{(af_1+bf_2)} = \int_0^1 \mathbb{E}_{\boldsymbol{\pi}(\boldsymbol{\epsilon}|\gamma\cdot\mathbb{1}_p)}\Big[[af_1(\boldsymbol{\epsilon}\circ\boldsymbol{x}\oplus\bar{\boldsymbol{\epsilon}}\circ\mathring{\boldsymbol{x}})+bf_2(\boldsymbol{\epsilon}\circ\boldsymbol{x}\oplus\bar{\boldsymbol{\epsilon}}\circ\mathring{\boldsymbol{x}})]\cdot(\frac{\boldsymbol{\epsilon}}{\gamma}-\frac{\bar{\boldsymbol{\epsilon}}}{1-\gamma})\Big]\,\mathrm{d}\gamma$$

$$= a\cdot\int_0^1 \mathbb{E}_{\boldsymbol{\pi}(\boldsymbol{\epsilon}|\gamma\cdot\mathbb{1}_p)}\Big[f_1(\boldsymbol{\epsilon}\circ\boldsymbol{x}\oplus\bar{\boldsymbol{\epsilon}}\circ\mathring{\boldsymbol{x}})\cdot(\frac{\boldsymbol{\epsilon}}{\gamma}-\frac{\bar{\boldsymbol{\epsilon}}}{1-\gamma})\Big]\,\mathrm{d}\gamma\,+$$

$$b\cdot\int_0^1 \mathbb{E}_{\boldsymbol{\pi}(\boldsymbol{\epsilon}|\gamma\cdot\mathbb{1}_p)}\Big[f_2(\boldsymbol{\epsilon}\circ\boldsymbol{x}\oplus\bar{\boldsymbol{\epsilon}}\circ\mathring{\boldsymbol{x}})\cdot(\frac{\boldsymbol{\epsilon}}{\gamma}-\frac{\bar{\boldsymbol{\epsilon}}}{1-\gamma})\Big]\,\mathrm{d}\gamma$$

$$= a\cdot\boldsymbol{\xi}^{(f_1)} + b\cdot\boldsymbol{\xi}^{(f_2)}$$

□

### A.2.4. SYMMETRY

In context of feature attribution, **Symmetry** states: given a function $f(\cdot)$ that is symmetric in two variables $x_i$ and $x_j$, the attribution scores of the two features satisfies $\xi_i = \xi_j$ when the explicand-baseline pair holds $x_i = x_j$ and $\mathring{x}_i = \mathring{x}_j$.

*Proof of Symmetry.* Similar to the proof of *Insensitivity*, the *Symmetry* of GEFA originates from the proxy gradient estimator. Let $x_i$ and $x_j$ denote the two symmetric features, their gradient estimators are:

$$\eta_{\alpha_i}(\boldsymbol{x}(\gamma\cdot\mathbb{1}_p)) = \mathbb{E}_{\boldsymbol{\pi}(\epsilon_i|\gamma)}\Big[\mathbb{E}_{\boldsymbol{\pi}(\boldsymbol{\epsilon}_{\backslash i}|\gamma\cdot\mathbb{1}_{p-1})}[f(\boldsymbol{\epsilon}\circ\boldsymbol{x}\oplus\bar{\boldsymbol{\epsilon}}\circ\mathring{\boldsymbol{x}})]\cdot(\frac{\epsilon_i}{\gamma}-\frac{\bar{\epsilon}_i}{1-\gamma})\Big]$$

$$\eta_{\alpha_j}(\boldsymbol{x}(\gamma\cdot\mathbb{1}_p)) = \mathbb{E}_{\boldsymbol{\pi}(\epsilon_j|\gamma)}\Big[\mathbb{E}_{\boldsymbol{\pi}(\boldsymbol{\epsilon}_{\backslash j}|\gamma\cdot\mathbb{1}_{p-1})}[f(\boldsymbol{\epsilon}\circ\boldsymbol{x}\oplus\bar{\boldsymbol{\epsilon}}\circ\mathring{\boldsymbol{x}})]\cdot(\frac{\epsilon_i}{\gamma}-\frac{\bar{\epsilon}_i}{1-\gamma})\Big]$$

Given the symmetry between $x_i$ and $x_j$, the inner expectations satisfy:

$$\mathbb{E}_{\boldsymbol{\pi}(\boldsymbol{\epsilon}_{\backslash i}|\gamma\cdot\mathbb{1}_{p-1})}[f(\boldsymbol{\epsilon}\circ\boldsymbol{x}\oplus\bar{\boldsymbol{\epsilon}}\circ\mathring{\boldsymbol{x}})] = \mathbb{E}_{\boldsymbol{\pi}(\boldsymbol{\epsilon}_{\backslash j}|\gamma\cdot\mathbb{1}_{p-1})}[f(\boldsymbol{\epsilon}\circ\boldsymbol{x}\oplus\bar{\boldsymbol{\epsilon}}\circ\mathring{\boldsymbol{x}})],\quad \text{when } \epsilon_i = \epsilon_j$$

It is not difficult to show that sampling of the two features following the same distribution given $x_i = x_j$ and $\mathring{x}_i = \mathring{x}_j$, which induces:

$$\eta_{\alpha_i}(\boldsymbol{x}(\gamma \cdot \mathbb{1}_p)) = \eta_{\alpha_j}(\boldsymbol{x}(\gamma \cdot \mathbb{1}_p))$$

Integrating the estimators having the same outputs along the symmetric path concludes the proof by showing:

$$\xi_i = \int_0^1 \eta_{\alpha_i}(\boldsymbol{x}(\gamma \cdot \mathbb{1}_p)) \, \mathrm{d}\gamma = \int_0^1 \eta_{\alpha_j}(\boldsymbol{x}(\gamma \cdot \mathbb{1}_p)) \, \mathrm{d}\gamma = \xi_j$$

$\square$

## A.3. Control Variate

### A.3.1. UNBIASEDNESS OF CONTROL VARIATE

To prove the unbiasedness of $\tilde{\boldsymbol{\xi}}$, we need to show $\tilde{\boldsymbol{\xi}} = \boldsymbol{\xi}$. Applying *Linearity*, we can rewrite $\tilde{\boldsymbol{\xi}}$ as:

$$\tilde{\boldsymbol{\xi}} = \boldsymbol{\xi}^{(f)} + \beta \cdot \boldsymbol{\xi}^{(h)} = \boldsymbol{\xi} + \beta \cdot \boldsymbol{\xi}^{(h)}$$

Now, the goal of the proof can be transformed to:

$$\tilde{\boldsymbol{\xi}} = \boldsymbol{\xi} \iff \boldsymbol{\xi}^{(h)} = \mathbb{0}_p$$

*Proof of Theorem 4.* The attribution of the control variate to the $i$-th feature is:

$$
\begin{aligned}
\xi_i^{(h)} &= \int_0^1 \mathbb{E}_{\boldsymbol{\pi}(\boldsymbol{\epsilon}|\gamma \cdot \mathbb{1}_p)}[h(\|\boldsymbol{\epsilon}\|_1) \cdot (\frac{\epsilon_i}{\gamma} - \frac{\bar{\epsilon}_i}{1-\gamma})] \, \mathrm{d}\gamma \\
&= \sum_{\boldsymbol{\epsilon} \in \{0,1\}^p : \epsilon_i = 0} \frac{\|\boldsymbol{\epsilon}\|_1!(p - \|\boldsymbol{\epsilon}\|_1 - 1)!}{p!} \cdot \left( h(\|\boldsymbol{\epsilon}\|_1 + 1) - h(\|\boldsymbol{\epsilon}\|_1) \right) && \text{(Theorem 2)} \\
&= \sum_{\|\boldsymbol{\epsilon}\|_1 = 0}^{p-1} \binom{p-1}{\|\boldsymbol{\epsilon}\|_1} \cdot \frac{\|\boldsymbol{\epsilon}\|_1!(p - \|\boldsymbol{\epsilon}\|_1 - 1)!}{p!} \cdot \left( h(\|\boldsymbol{\epsilon}\|_1 + 1) - h(\|\boldsymbol{\epsilon}\|_1) \right) \\
&= \sum_{\|\boldsymbol{\epsilon}\|_1 = 0}^{p-1} \frac{1}{p} \cdot \left( h(\|\boldsymbol{\epsilon}\|_1 + 1) - h(\|\boldsymbol{\epsilon}\|_1) \right) \\
&= \frac{1}{p} \cdot \left( h(p - 1 + 1) - h(0) \right) && \text{(Telescoping series)} \\
&= 0
\end{aligned}
$$

The zeroness of feature attribution $\xi_i^{(h)}$ concludes the proof:

$$\xi_i^{(h)} = 0, \ \forall i \in \{1, 2, \ldots, p\} \implies \boldsymbol{\xi}^{(h)} = \mathbb{0}_p$$

$\square$

While constructing the control variate for GEFA, we first initialize it as $h(\|\boldsymbol{\epsilon}\|_1) = \|\boldsymbol{\epsilon}\|_1/p$ based on Assumption 3. To strictly follow the property of unbiasedness, the above analysis derives an additional requirement for the control variate, namely:

$$h(p) = h(0)$$

Integrating the constraint into the control variate delivers the function stated in (9). In addition to the selected control variate, Theorem 3 applies to the broader group of functions, which depends solely on $\|\boldsymbol{\epsilon}\|_1$ and at the same time satisfies $h(p) = h(0)$. When there are further assumptions to make on the target function, the shape of $h(\cdot)$ can be fine-tuned for a stronger covariance in relation to $f(\cdot)$.

### A.3.2. OPTIMALITY OF CONTROL VARIATE COEFFICIENT

Next, we show the variance reduction effect of the control variate is optimized when:

$$\beta^* = \text{Cov}(f, h)/\text{Var}(h)$$

where the optimal choice of the weighting term is denoted as $\beta^*$.

*Proof of Optimality of $\beta^*$.* Denoting the variance of a gradient estimator for a feature $x_i$ as $\text{Var}(\xi_i)$, the variance of the estimator after the introduction of a control variate is:

$$\text{Var}(\tilde{\xi}_i) = \text{Var}(\xi_i) + \beta^2 \text{Var}(\xi_i^{(h)}) - 2\beta \cdot \text{Cov}(\xi_i, \xi_i^{(h)})$$

The optimal variance reduction effect for $\xi_i$ is achieved when:

$$\beta = \text{Cov}(\xi_i, \xi_i^{(h)})/\text{Var}(\xi_i^{(h)}) \tag{12}$$

Alternative to a feature-specific optimal value, we are also interested in a single value for $\beta$ that maximizes the overall variance reduction effect. To acquire the overall optimum, we first expand the covariance in (12):

$$\text{Cov}(\xi_i, \xi_i^{(h)}) = \mathbb{E}[\xi_i \cdot \xi_i^{(h)}] - \mathbb{E}[\xi_i] \cdot \mathbb{E}[\xi_i^{(h)}]$$
$$= \mathbb{E}_{\alpha_i}\left[\mathbb{E}_{\epsilon_i}[f(\boldsymbol{z}) \cdot h(\|\boldsymbol{\epsilon}\|_1) \cdot (\nabla_{x_i} \log \pi(\epsilon_i|\alpha_i))^2]\right] - \mathbb{E}[\xi_i] \cdot 0 \qquad \text{(Unbiasedness of } \boldsymbol{\xi}^{(h)})$$

Please note that we omit the distribution that $\alpha_i$ and $\epsilon_i$ should follow as it does not affect the result of the derivation. For high-dimensional input, the functions $f(\cdot)$ and $h(\cdot)$ have trivial dependencies on a specific feature $x_i$:

$$\text{Cov}(\xi_i, \xi_i^{(h)}) \approx \mathbb{E}_{\alpha_i}\left[\mathbb{E}_{\epsilon_i}[f(\boldsymbol{z}) \cdot h(\|\boldsymbol{\epsilon}\|_1)]\right] \cdot \mathbb{E}_{\alpha_i}\left[\mathbb{E}_{\epsilon_i}[(\nabla_{x_i} \log \pi(\epsilon_i|\alpha_i))^2]\right]$$

Similarly, the variance of the control variate estimator can be written as:

$$\text{Var}(\xi_i^{(h)}) \approx \mathbb{E}_{\alpha_i}\left[\mathbb{E}_{\epsilon_i}[h(\|\boldsymbol{\epsilon}\|_1)^2]\right] \cdot \mathbb{E}_{\alpha_i}\left[\mathbb{E}_{\epsilon_i}[(\nabla_{x_i} \log \pi(\epsilon_i|\alpha_i))^2]\right]$$

Putting together yields the overall optimal value $\beta^*$:

$$\beta^* = \frac{\mathbb{E}_{\alpha_i}\left[\mathbb{E}_{\epsilon_i}[f(\boldsymbol{z}) \cdot h(\|\boldsymbol{\epsilon}\|_1)]\right] \cdot \mathbb{E}_{\alpha_i}\left[\mathbb{E}_{\epsilon_i}[(\nabla_{x_i} \log \pi(\epsilon_i|\alpha_i))^2]\right]}{\mathbb{E}_{\alpha_i}\left[\mathbb{E}_{\epsilon_i}[h(\|\boldsymbol{\epsilon}\|_1)^2]\right] \cdot \mathbb{E}_{\alpha_i}\left[\mathbb{E}_{\epsilon_i}[(\nabla_{x_i} \log \pi(\epsilon_i|\alpha_i))^2]\right]}$$

$$= \frac{\mathbb{E}_{\alpha_i}\left[\mathbb{E}_{\epsilon_i}[f(\boldsymbol{z}) \cdot h(\|\boldsymbol{\epsilon}\|_1)]\right] - 0}{\mathbb{E}_{\alpha_i}\left[\mathbb{E}_{\epsilon_i}[h(\|\boldsymbol{\epsilon}\|_1)^2]\right] - 0}$$

$$= \frac{\mathbb{E}_{\alpha_i}\left[\mathbb{E}_{\epsilon_i}[f(\boldsymbol{z}) \cdot h(\|\boldsymbol{\epsilon}\|_1)]\right] - \mathbb{E}_{\alpha_i}\left[\mathbb{E}_{\epsilon_i}[f(\boldsymbol{z})]\right] \cdot \overbrace{\mathbb{E}_{\alpha_i}\left[\mathbb{E}_{\epsilon_i}[h(\|\boldsymbol{\epsilon}\|_1)]\right]}^{=0}}{\mathbb{E}_{\alpha_i}\left[\mathbb{E}_{\epsilon_i}[h(\|\boldsymbol{\epsilon}\|_1)^2]\right] - 0}$$

$$= \text{Cov}(f, h)/\text{Var}(h)$$

$$\square$$

Taking the optimal $\beta^*$, the variance reduction effect depends on the square of the covariance between $f(\cdot)$ and $h(\cdot)$, which motivates Assumption 3:

$$\text{Var}(\xi_i) - \text{Var}(\tilde{\xi}_i) = \text{Cov}(f, h)^2$$

## A.4. Equivalence to IG

*Proof of Theorem 5.* To complete the proof, we first show that both GEFA and IG produce marginal contributions along edge paths.

Recalling that an edge path always moves from one vertex $z_S$ to an adjacent vertex that differs $z_{S \cup \{i\}}$ in only the $i$-th feature along edges, the goal is simplified to prove that they are calculators of the marginal contribution conditioned on the presence of features $\{x_j | j \in S\}$ for each segment of a path. For the $i$-th segment on an edge path with $S$ denoting the preceding vertices, IG produces:

$$\xi_i^{\text{IG}} = \int_{z_S}^{z_{S \cup \{i\}}} \frac{\partial f(x)}{\partial x_i} \, dx$$
$$= f(z_{S \cup \{i\}}) - f(z_S)$$

As the path for GEFA is created in the proxy space, we denote the two proxy vertices on the $i$-th segment by $x(\alpha_S)$ and $x(\alpha_{S \cup \{i\}})$ for preciseness. The notation $\alpha_S$ is analogous to $z_S$, which represents:

$$\alpha_i = \begin{cases} 1 & \text{if } i \in S \\ 0 & \text{if } i \notin S \end{cases}$$

When following the same permutation order, GEFA produces the same marginal contribution as IG for the $i$-th segment:

$$\xi_i^{\text{GEFA}} = \int_{\alpha_S}^{\alpha_{S \cup \{i\}}} \mathbb{E}_{\pi(\epsilon_i | \alpha_i)}[f(z) \cdot (\frac{\epsilon_i}{\alpha_i} - \frac{\bar{\epsilon}_i}{1 - \alpha_i})] \, d\alpha$$
$$= f(z_{S \cup \{i\}}) - f(z_S)$$
$$\Leftrightarrow \xi_i^{\text{IG}}$$

Please note that, for GEFA, the only feature value in $z$ that may vary during the sampling on the $i$-th segment is $z_i$. The remaining features are deterministic as their corresponding proxy variables are either 0 or 1 depending on whether they have been visited in the preceding vertices $S$, namely to take either the baseline or explicand value with a hundred percent probability.

As both explainers deliver marginal contributions along edge paths, the claim in Theorem 5 becomes obvious as it describes the typical computation of Shapley Values. □

# B. Detailed Experimental Setting

## B.1. Experimental Environment

The competitors, including the proposed method, were implemented using Python 3.11.2 with standard packages. The primary packages were Numpy 1.26.4, PyTorch of version 2.5.0, and Torchvision 0.20.0. The CUDA version was 12.2 for GPU support. All experiments were conducted on a machine operated by Debian 11 with the following specifications:

- Processor: Intel i9-10980XE, 18 cores

- Memory: 32GB DDR4

- GPU: NVIDIA RTX A5500, 24GB

## B.2. Tested Models

The experiment section used four models for the evaluation of the attribution methods, namely BERT, Llama3, InceptionV3, and Vision Transformer. These models are open-source with publicly available pretrained versions, ensuring the reproducibility of our experimental results. The specific versions of the pretrained models are listed below:

- BERT: `bert-base-uncased` released on Hugging Face[4]

- Llama3: `Llama3.2-3B-Instruct` released on Hugging Face[5]

---

[4]https://huggingface.co/google-bert/bert-base-uncased
[5]https://huggingface.co/meta-llama/Llama-3.2-3B-Instruct

Table 3: Model performance on corresponding dataset

| Dataset
Model | Amazon
BERT | STS-2
Llama3.2-3B | QNLI
Llama3.2-3B | ImagetNet
InceptionV3 | ImageNet
ViT |
|---|---|---|---|---|---|
| Acc. (%) | 96.63 | 92.55 | 80.26 | 77.29* | 81.07* |

*Top-1 accuracy reported by the model provider.

- InceptionV3: `inception_v3` with the pretrained weights `IMAGENET1K_V1` released on PyTorch[6]

- Vision Transformer: `vit-b-16` with the pretrained weights `IMAGENET1K_V1` released on PyTorch[7]

While most models are specified for classification tasks, Llama3 is a general-purpose LLM. To facilitate the focus on the standard feature attribution problem, we configure the behavior of Llama3 through prompt engineering and use it as a zero-shot learner for STS-2 and QNLI.

**STS-2 Prompt**

```
You are a sentiment classifier trained on movie reviews. Your task is to identify the
    sentiment of the given input. If the sentiment of the text is negative, output 0.
    If the sentiment is positive, output 1.
Input: {Input};
Output:
```

**QNLI Prompt**

```
You are a highly accurate classifier for question-sentence pairing. Your task is to
    determine whether the provided sentence answers the given question. If the sentence
     answers the question, output 1. If the sentence does not answer the question,
    output 0.
Input:
Question: {Input Part 1}
Sentence: {Input Part 2};
Output:
```

The placeholders in the task-specific prompts are later filled with content tokens from the concrete inputs. The above task-specific prompts guide the language model to put its answer as either "0" or "1" in the next predicted token. This setting circumvents the analysis of auto-regressive text generation and reduces the total number of output nodes from the vocabulary size of $128,000$ to 2. Based on this setup, the task of the explainers is specified as determining model attributions to the content tokens. The template tokens receive zero attributions because the template prompts remain unchanged during the explanation process, with the template tokens assigned identical values for both the explicand and the baseline.

The accuracies of the models are presented in Table 3. The performances of InceptionV3 and ViT are sourced from the PyTorch website, as no additional fine-tuning steps were conducted on the two pretrained models.

## B.3. Competitors and Implementation Details

### B.3.1. WHITE-BOX EXPLAINERS

The two white-box approaches derive explanations based on exact gradient measurements. Specifically, *VG* interprets raw gradients directly as explanations, and *IG* integrates gradients along a straightline path between the explicand and the baseline. These approaches are closely connected to the proposed method due to the shared concept of utilizing gradients. While implementing gradient-based solutions is straightforward with existing tools, special care has been taken for explaining text classifiers. When processing textual inputs, gradients for an explicand are not directly available due to the non-differentiable nature of discrete features. As a workaround, we follow the common practice of first computing attribution scores for the embeddings, which represent the discrete input features in the continuous embedding space, and then aggregating these scores into token-level attributions. Formally, let $\boldsymbol{x}$ be a textual explicand with $p$ tokens and $\boldsymbol{e} \in \mathbb{R}^{p \times q}$

---

[6]https://pytorch.org/vision/main/models/inception.html
[7]https://pytorch.org/vision/main/models/vision_transformer.html

represent the corresponding representation in the embedding space of size $q$. The attribution of each embedding element $\xi_{e_{ij}}$ is derived following the standard back-propagation process for continuous variables. The token-level attribution $\xi_{x_i}$ is then determined as the sum of attributions over the corresponding embedding vector:

$$\xi_{x_i} = \sum_{j=1}^{q} \xi_{e_{ij}}$$

Summing embedding attributions appears intuitive, but it can be easily proved that this approach preserves the promising properties of IG.

### B.3.2. Black-box Explainers

The black-box competitors are SHAP and GEEX. The former is connected to the proposed method through Shapley Values, and the latter is a recent attempt at applying gradient estimation in feature attribution, which inspired this work. There are two variants of SHAP considered in the experiment section. *KSHAP* is a Shapley Value estimator based on weighted linear regression, and *PSHAP* is a sampling-based estimator that computes Shapley Values recursively through a hierarchy of features (Chen et al., 2023). To avoid reinventing the wheel, the official implementations[8] were used for evaluating the two SHAP variants, and the code open-sourced by Cai & Wunder (2024) was used for GEEX.

### B.3.3. Implementation Details of GEFA

In addition to the control variate, we employed *mask smoothing* (Cai & Wunder, 2024) and *antithetic sampling* for variance reduction during the implementation of GEFA. The two techniques proposed in previous work were adapted to fit into the context of proxy gradient estimation.

Masking smoothing is a post-processing technique for mask generation, typically applied when explaining image classifiers. Following the intuition that adjacent pixels form low-level patterns, mask smoothing processes masks with a low-pass filter, softly grouping spatially close pixels and consequently reducing the feature space. By applying similar changes to adjacent pixels, mask smoothing enhances the possibility of masking out local patterns, thereby exposing model sensitivities to the absence of certain features. The sampling process of GEFA incorporates mask smoothing when explaining InceptionV3 and ViT. For each position indicator $\gamma$ on the proxy path, masks are initialized with a uniform distribution $\varepsilon \sim \mathcal{U}_{[0,1]}$. After the initialization of uniform noises, the number of present features is determined by the threshold $\gamma$, denoted as $k = \|\varepsilon > \gamma\|_1$. Subsequently, the smoothed masks $\dot{\varepsilon}$ are generated by applying a low-pass filter $w$ to the uniform random masks, softly grouping nearby pixels for better coverage of low-level patterns. It is important to note that the smoothed mask values no longer follow the uniform distribution; therefore, the relative ranking of mask values is considered during binarization instead of their numerical values. Specifically, the smoothed mask is binarized by setting the top-$k$ units to positive, thereby delivering the target binary mask $\epsilon$[9] for GEFA. Algorithm 2 provides an overview of the sampling process.

---

**Algorithm 2** Smoothing Enhanced Mask Sampling

---

**Input:** $\gamma$: path position indicator; $w$: low-pass filter
**Output:** $\epsilon$: binary mask;

1: $\varepsilon \sim \mathcal{U}_{[0,1]}^p, \epsilon = \mathbb{0}_p$          *# Mask initialization*
2: $k = \|\varepsilon > \gamma\|_1$          *# Number of present features*
3: $\dot{\varepsilon} = \varepsilon * w$          *# Mask smoothing*
4: $\rho = \mathrm{argsort}(\dot{\varepsilon})$          *# Indices of sorted values*
5: **for** $i$ in $\rho[:k]$ **do**
6:     $\epsilon_i = 1$          *# Set top-k features as present*
7: **end for**
8: **return** $\epsilon$

---

For each mask $\varepsilon \in \mathbb{R}^p$ in a continuous space, antithetic sampling takes its antithesis $-\varepsilon$ for the estimation, namely creating an antithetic query pair $\{x + \varepsilon, x - \varepsilon\}$. However, the binary nature of masks $\epsilon \in [0,1]^p$ in proxy gradient estimation prohibits the direct application of this sampling strategy. To address this, we first define the antithesis of a binary mask

---

[8]https://shap.readthedocs.io/en/latest/#
[9]Please note that the different formats, $\varepsilon$ and $\epsilon$, are used to distinguish between continuous and binary masks.

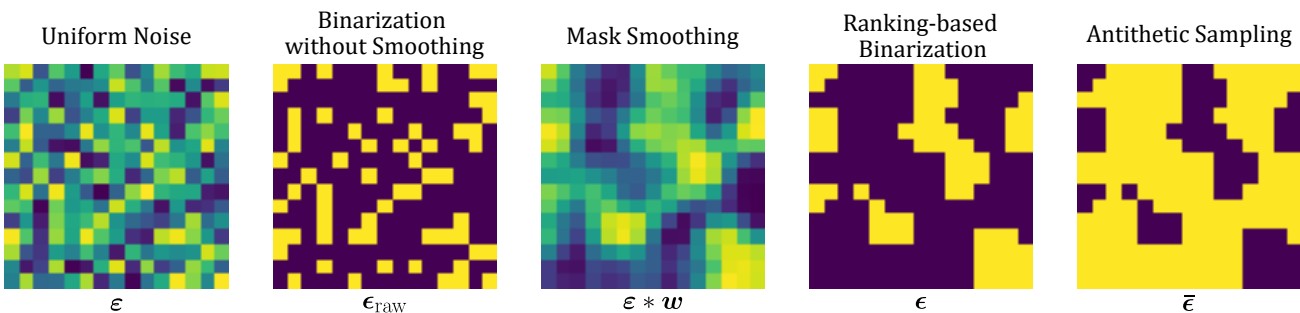

Figure 3: Visualization of masks at different phases of the sampling process. Binary masks $\epsilon$ and $\bar{\epsilon}$ derived from the smoothed noise are used to generate queries.

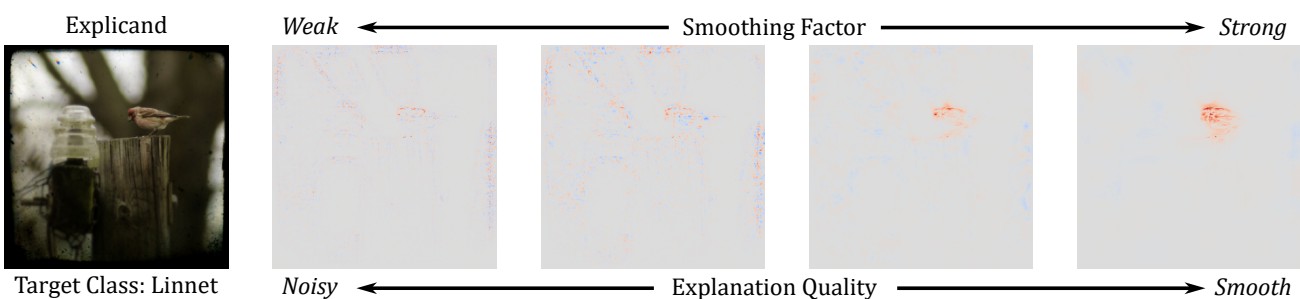

Figure 4: Visualization of mask smoothing impacts on explanation quality

as $\bar{\epsilon}$, which flips the presence/absence states represented in the mask. Second, the antithesis of $\epsilon \sim \boldsymbol{\pi}(\cdot|\gamma \cdot \mathbb{1})$ follows a distribution $\boldsymbol{\pi}(\cdot|(1-\gamma) \cdot \mathbb{1})$, which must be recorded for the correct estimation of $\boldsymbol{\xi}$. Finally, an antithetic query pair in the context of GEFA is defined by:

$$\left\{ (\epsilon \circ \boldsymbol{x} \oplus \bar{\epsilon} \circ \mathring{\boldsymbol{x}}, \gamma), (\bar{\epsilon} \circ \boldsymbol{x} \oplus \epsilon \circ \mathring{\boldsymbol{x}}, 1-\gamma) \right\}$$

### B.3.4. TIME COMPLEXITY

The computational expenses of white-box and black-box approaches arise from different sources due to their distinct accessibility assumptions. Let $\mathcal{O}(\mathcal{M})$ denote the model complexity, the time cost of gradient-based approaches is $\mathcal{O}(s \cdot \lambda \mathcal{M})$, where $\lambda \mathcal{M}$ indicates the cost for gradient measurement and $s$ is the number of measurements required for deriving the final explanation. For example, $s$ corresponds to the number of interpolation steps in IG. On the other hand, the computational cost of black-box approaches is composed of two primary factors: query generation and model inference.

This section mainly focuses on the complexity of GEFA and its comparison to other black-box explainers functioning on a query basis. Given a total budget of $n$ queries in a $p$-dimensional feature space, the cost of query generation for GEFA is $\mathcal{O}(np)$. The overall complexity of GEFA combines the generation cost and model complexity, expressed as $\mathcal{O}(np + n\mathcal{M})$, which is equivalent to the complexity of GEEX. Between the two factors, the total time cost in practice is dominated by $\mathcal{O}(n\mathcal{M})$, as GEFA completes mask construction upon initialization. The pre-constructed masks accelerate the query generation process, thereby reducing the relevant cost. For PSHAP and KSHAP, the complexity increases to $\mathcal{O}((n + \tau) \cdot p + n\mathcal{M})$, where the additional cost $\tau$ arises from explicand-specific feature space partition (PSHAP) and linear regression (KSHAP). The impact of the additional cost becomes non-trivial when the dimensionality $p$ increases. Table 4 reports the explanation time costs of the competitors. For the group of black-box explainers, when given the same query budget, the computational expenses align with the above analysis.

Table 4: Time cost (s) per explicand

| Model | VG | IG | KSHAP | PSHAP | GEEX | GEFA |
|---|---|---|---|---|---|---|
| BERT | 0.04 | 0.20 | 1.33 | 0.75 | - | 0.82 |
| Llama3 | 0.19 | 1.40 | 5.53 | 5.09 | - | 5.42 |
| InceptionV3 | 0.06 | 1.53 | - | 15.63 | 4.57 | 4.91 |
| ViT | 0.04 | 1.11 | - | 23.54 | 16.17 | 16.97 |

## C. Discussion on Evaluation Scheme

### C.1. Validity of Evaluation via Deletion

This work adopts evaluation via deletion to assess explanation quality, allowing quantitative comparison among the competitors. For simplicity, the adopted evaluation strategy is referred to as the *deletion scheme* in the rest of the discussion. The employment of the deletion scheme spans from the early stages of explainability research (Samek et al., 2016; Montavon et al., 2018) to the most recent studies (Cai & Wunder, 2024; Muzellec et al., 2024). It offers static environments for efficient explanation quality assessment.

One major concern about the validity of the deletion scheme is the issue of distribution shift. During the evaluation process, input features are recursively removed from an explicand, resulting in manipulated copies that deviate from the original data manifold. These copies with artifacts introduce distribution shift as another potential source of model performance degradation, in addition to the intended effect of feature absence. To alleviate the concern, we intentionally include random removal as a reference for the effectiveness of explanation-guided deletion. By excluding explanatory information from the manipulation process, the figures obtained through random removal reflect the extent of degradation by distribution shift alone. The experimental results in Table 1 and Table 2 demonstrate that most of the explainers significantly outperform random removal across various settings, highlighting the effectiveness of the derived explanations. Moreover, the random-level performance of VG matches the analysis of its limitation in previous work (Sundararajan et al., 2017). The alignment between the empirical results and theoretical analysis further emphasizes the validity of the adopted evaluation scheme. While the experimental results, by themselves, mitigate concerns about the validity of the deletion scheme, we provide additional results in Appendix C.3 to demonstrate its consistency with the retraining scheme. The retraining scheme (Hooker et al., 2019), an evaluation scheme designed to overcome the out-of-distribution issue, is elaborated on in the following section.

### C.2. Distortion in the Retraining Scheme

Hooker et al. (2019) highlighted the issue of distribution shift in explanation evaluation, expressing concerns about unexpected model behaviors triggered by artifacts. As an alternative to the traditional deletion scheme, they proposed a retraining scheme known as r̲emo̲ve a̲nd r̲etraining (ROAR). This approach involves removing a proportion of features with the highest attribution scores for each instance in the dataset, followed by retraining the model on the manipulated dataset. The performance of the retrained model is then used as an indicator of the explainer's effectiveness. The described scheme poses the question:

$Q_1$: *"Does the tested explanation method identify all task-relevant features?"*

Following the idea of retraining, ROAR's assessments reported that many popular attribution methods "*are not better than a random designation of feature importance*". This observation directly raises concerns about the validity of the deletion scheme. However, this conclusion drawn from ROAR's assessments conflicts with the theoretical guarantees and the widely approved effectiveness of many test approaches, which prompted our further investigation. Upon reproducing ROAR's assessments, we identified the root cause of the misalignment between evaluations from different perspectives to be **residual information**, which *distorts* the results of the retraining scheme. With a justified adjustment to the retraining scheme, we found that the two evaluation strategies yield consistent results.

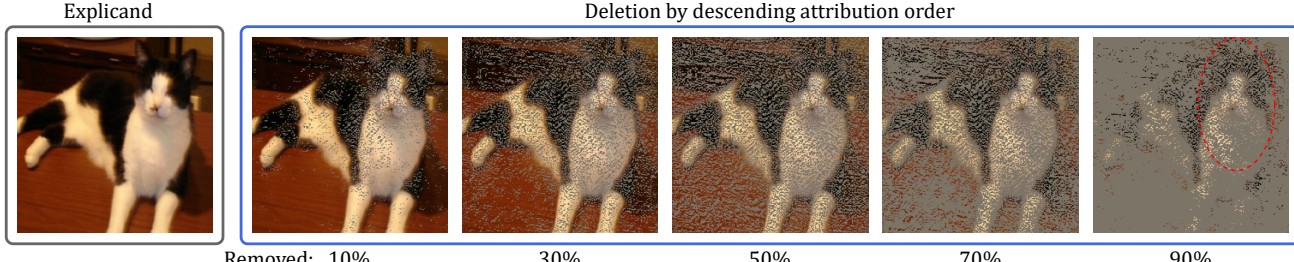

Figure 5: Residual information in the retraining scheme: An example of feature deletion by attribution order. Negatively attributed features retained after manipulation cause information leakage, distorting evaluation results. The red dashed line highlights residual information in the manipulated copy with $90\%$ of features removed.

### THE RESIDUAL INFORMATION

The *sign* issue is a major source of residual information, referring to information leakage caused by negatively attributed features retained in the retraining scheme. In the context of feature attribution, a positive attribution score indicates a positive contribution to the prediction result, whereas a negative score represents, rather than irrelevance, a negative association with the decision. The origin of negative attributions is complex, influenced by multiple factors such as the baseline choice and the associations between features and the prediction function. When features are removed in descending order of attribution scores, negatively attributed features inevitably remain in the manipulated dataset for retraining. The failure to remove these "negative" features preserves task-relevant information, which the model can reorganize during retraining to improve accuracy.

A qualitative example of the residual "negative" information in the retraining scheme is illustrated in Figure 5. The *sign* issue also explains the effective manipulation of SG-SQ and VarGrad, as reported by Hooker et al. (2019). Rather than a miracle of explainer ensembling, the two variants of gradient-based explanation provide unsigned attributions, ensuring a more comprehensive removal of informative features. It is noteworthy that, unlike the retraining scheme, the traditional deletion scheme is insensitive to residual information. Without retraining, the negative associations of the retained features are fixed and will not be updated in the consistent environment offered by the traditional deletion scheme.

In addition to the sign issue, *feature redundancy* poses another challenge to the implicit assumption made by ROAR. The removal of task-relevant features, as required by $Q_1$, depends on both the model and the explainer. By expecting significant performance degradation after retraining, ROAR assumes that the to-be-explained model has learned all task-relevant features and interprets them in a way that truthfully reflects their relevance. However, this assumption is an over-qualified requirement for machine learning models. Previous research (Lapuschkin et al., 2019; Geirhos et al., 2020) has shown that the inference of a trained model can be dominated by a subset of features, leaving many relevant features under-attributed. Without any guarantee of comprehensive feature capturing, redundancy can lead to the leakage of task-relevant information after dataset manipulation, thereby introducing distortion into the evaluation results.

### C.3. Corrected Retraining Scheme

Given the various sources of residual information, a direct cleaning of the manipulated dataset is unfeasible, as it is extremely difficult – if not impossible – to distinguish between model-sourced feature omissions and explainer-sourced. To efficiently address the assessment distortion in ROAR, we propose that, instead of removing features, the top-ranked features should be retained for retraining. The "keep and retrain" (KEAR) approach reframes the evaluation question as:

$Q_2$: *"Does the tested explanation method effectively identify relevant features?"*

An effective explanation method should capture the relevant information learned by the target model, resulting in less performance degradation of the retrained model with the same portion of retained information. To verify the above discussion, we conducted experiments following the retraining scheme in both text and image settings:

- The pretrained BERT is fine-tuned on the Amazon dataset for text classification

Table 5: Evaluation schemes comparison on BERT

| Competitor | In Acc. (%) | | nAOPC $\uparrow$ |
|:---:|:---:|:---:|:---:|
| | ROAR $\downarrow$ | KEAR $\uparrow$ | |
| VG | **77.92** | 79.69 | 18.23 |
| IG | 96.51 | 96.47 | 66.77 |
| KSHAP | 92.57 | 95.93 | 60.14 |
| PSHAP | 93.78 | 96.28 | 65.92 |
| GEFA | 94.86 | **96.56** | **73.66** |
| Random | 66.48 | | 19.08 |

$\downarrow$: lower is better;  $\uparrow$: higher is better

Table 6: Evaluation schemes comparison on EfficientNet

| Competitor | In Acc. (%) | | nAOPC $\uparrow$ |
|:---:|:---:|:---:|:---:|
| | ROAR $\downarrow$ | KEAR $\uparrow$ | |
| VG | 75.75 | 73.35 | 38.87 |
| IG | 77.20 | 89.60 | **40.82** |
| PSHAP | 79.25 | 84.30 | 39.56 |
| GEEX | **71.05** | 78.60 | 38.39 |
| GEFA | 82.35 | **89.95** | 40.79 |
| Random | 71.30 | | 35.07 |

- The pretrained EfficientNet-B0[10] is fine-tuned on the Cats vs. Dogs dataset (Elson et al., 2007)

For both settings, copies of the corresponding datasets were created with explanation-guided manipulation and then used for retraining the target model to assess explanation quality. Without losing generality, we adopted a lightweight model for image classification and downsampled the dataset into $2000/400/400$ partitions for training, validation, and test sets to ensure feasibility and efficiency. EfficientNet-B0 achieved an accuracy of $99.40\%$ on the downsampled dataset after fine-tuning. Details about the fine-tuned BERT model are given in Appendix B.2. Based on the fine-tuned models, we derived explanations for all datasets with the competitors selected in Section 5, namely, *VG*, *IG*, *PSHAP*, *KSHAP*, *GEEX*, and GEFA. Features for each instance were ranked in descending order according to their attribution scores. Similar to the deletion scheme, we adopted random removal as a reference to highlight the effectiveness of the competitors.

In the retraining tests, the top $90\%$ of features were **removed** for ROAR to create the manipulated datasets, whereas the top $10\%$ of features were **retained** for KEAR. To minimize the potential impact of randomness during the training process, we independently retrained five models on each manipulated dataset and reported the averaged accuracies. It is noteworthy that lower retraining accuracies indicate better explanation quality in the removal tests, whereas higher accuracies reflect superior explainer performance in KEAR. Results from different evaluation schemes are presented in Table 6. For random removal, the same figures are reported for both retraining settings because of the identical proportion of remaining features, i.e. $10\%$.

According to ROAR, all explanation methods exhibit minor manipulation impacts due to the previously discussed residual information, failing to excel random removal. This observation closely aligns with the finding in the original ROAR paper (Hooker et al., 2019). In contrast, by addressing the distortion caused by residual information, KEAR offers a more faithful assessment of explanation quality. The success of the explainers in identifying the most informative features results in relatively high classification accuracy, even with only $10\%$ of features retained. In addition to the retraining tests, the last column of Table 6 presents the nAOPC scores obtained following the deletion scheme. While the metrics employed by the two schemes differ in scale, leading to difficulties in direct numerical comparisons, the consistency in relative rankings within each test highlights nAOPC as a valid metric. The KEAR results, alongside nAOPC scores, demonstrate that the retraining scheme and the deletion scheme are parallel evaluation options rather than conflicting approaches.

## D. Additional Experimental Results

### D.1. Sensitivity to Query Budget

Section 5 quantitatively compared the performances of SOTA feature attribution methods with an empirically selected query budget. To better understand the sensitivity of the proposed method to the query budget $n$ and its convergence behavior as an estimator, we performed a grid search evaluation over $n$ to assess its impact on explanation quality. The solid blue lines in Figures 6a and 6b show the nAOPC scores of GEFA on *BERT-Amazon* and *InceptionV3-ImageNet*, respectively, under various query budgets. The query budget scale is indicated by the x-axis scales at the top of each diagram. Since the search space of proxy gradient estimation depends on feature space dimensionality, the scale of query budgets varies between the two cases, ranging from $100$ to $5k$ for BERT (left) and from $1k$ to $50k$ for InceptionV3 (right). As the query budget increases, the convergence behavior of GEFA is consistent in both cases. The increase in explanation quality shows

---
[10]https://pytorch.org/vision/main/models/efficientnet.html

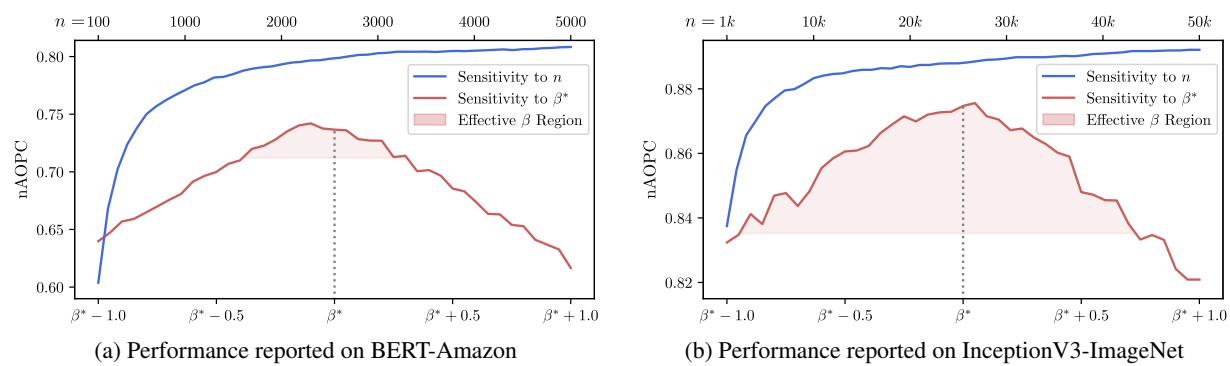

(a) Performance reported on BERT-Amazon    (b) Performance reported on InceptionV3-ImageNet

Figure 6: Sensitivity analysis of the query budget $n$ and the control variate coefficient $\beta$. Explanation quality (measured by nAOPC, y-axis) is shown against $n$ (blue line, top x-axis scale) and $\beta$ (red line, bottom x-axis scale). The shaded red region highlights the performance gain achieved with a properly weighted control variate.

an accelerated pattern with a low query budget, and tends to plateau toward the higher end of the query budget.

### D.2. Sensitivity to Control Variate Coefficient

The grid search evaluation was also performed to analyze the optimality of the control variate coefficient $\beta^*$. Different from the query budget, which is constant during each test, the value of $\beta^*$ is explicand-specific. The computation of $\beta^*$ depends on the local behavior of $f(\cdot)$ specified by the explicand-baseline pair $(\boldsymbol{x}, \mathring{\boldsymbol{x}})$, which varies across different explicands. Taking the variability into account, $\beta$ was set by, instead of absolute values, relative values to $\beta^*$ during the evaluation. Specifically, for both settings, we tested the performance of GEFA with $\beta$ ranging from $[\beta^* - 1, \beta^* + 1]$, shown by the x-axis scales at the bottom of the diagrams. The query budget was fixed at $500$ and $5k$ for BERT-Amazon and InceptionV3-ImageNet, respectively.

The solid red lines in Figures 6a and 6b illustrate GEFA's performance with varying values of $\beta$. Generally, explanation performance peaks at $\beta^*$ and degrades toward both positive and negative deviations from the estimated optimum. Although slight discrepancies are observed between the best explanation quality and the estimated $\beta^*$, the overall performance trend aligns with the theoretical analysis and supports the optimality of $\beta^*$. The bias in the estimated optimum originates from compromises with the limited query budget. To avoid imposing additional query burdens, the coefficient estimation reuses observations from the explanation derivation process, which can introduce small biases in the estimated optimum (Mohamed et al., 2020). Practically, this bias diminishes quickly as the number of observations increases, explaining the relatively smaller divergence in the image classifier case due to the larger query budget. Despite the minor biases, the application of the control variate induces considerable performance gains almost free of charge, especially without increasing the query expense. Therefore, applying the control variate with the estimated coefficient $\beta^*$ should be considered a valuable practical tool to enhance explanation quality.

Another important observation is the larger performance gain region in Figure 6b, where the control variate exhibits a more significant impact on improving explanation quality. This observation is consistent with the greater performance gains observed in explaining image classifiers compared to text classifiers, as reported in Tables 1 and 2. The difference in the effectiveness of the control variate is further studied in the following section, which identifies the larger covariance between $f(\cdot)$ and $h(\cdot)$ as the origin of the enhanced impact.

### D.3. Control Variate Impact and Function Correlation

The analysis in Appendix A.3 concludes that the variance reduction effect brought by the control variate is determined by its covariance with the target function:

$$\text{Var}(\xi_i) - \text{Var}(\tilde{\xi}_i) = \text{Cov}(f, h)^2 \tag{13}$$

The conclusion offers a reasonable explanation for the larger performance improvements of GEFA when explaining InceptionV3 and ViT.

To verify this point, we collected and summarized model prediction confidences at different ratios of present features. Figure 7a illustrates the correlations between model outcomes and the ratios of present features across various test settings.

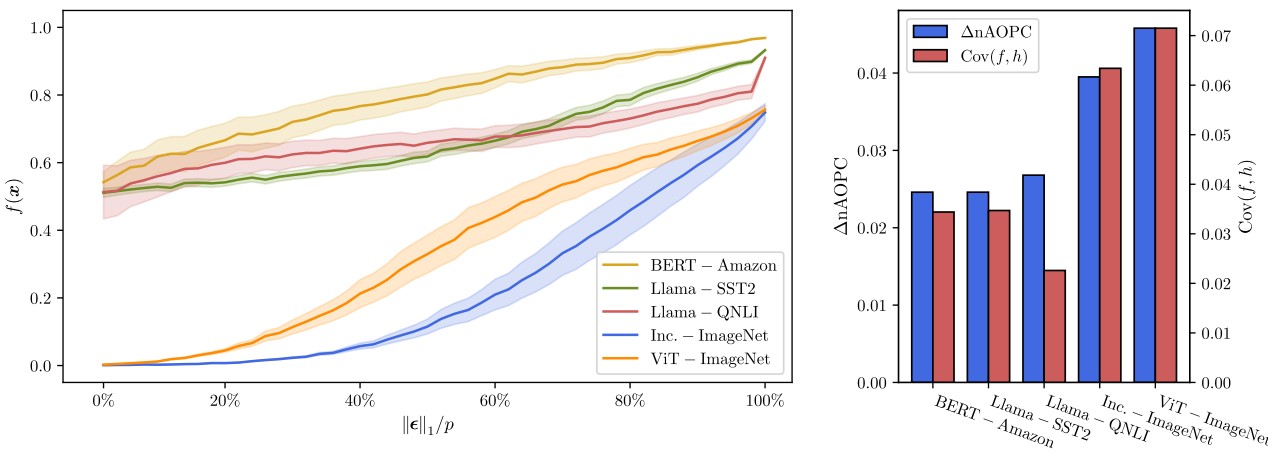

(a) Relationship between model outcome and ratio of present feature    (b) Performance improvements and covariances

Figure 7: Explanation quality improvement and Covariance between $f(\cdot)$ and $h(\cdot)$ and

The solid lines represent the average target function behavior, while the shaded regions indicate the variance. According to the figure, model prediction outcomes in all tested settings exhibit a positively correlation with the ratios of present features, supporting our argument for the validity of Assumption 3. Among them, the two image classifiers yield more pronounced performance sensitivity to $\|\boldsymbol{\epsilon}\|_1$.

Figure 7b visually compares the performance gains with the covariances $\mathrm{Cov}(f, h)$, estimated based on the averaged behaviors demonstrated in Figure 7a. Consistent with the analysis of the variance reduction effect as concluded by (13), the covariances between the target function and the control variate are more significant in the two image settings, leading to greater improvements in explanation quality. Additionally, according to Appendix A.3.2, the effective range of $\beta$ is given by:

$$\beta \in (0, \frac{2\mathrm{Cov}(f, h)}{\mathrm{Var}(h)})$$

With the variance of the designed control variate expressed in closed form as $\mathrm{Var}(h) = \frac{1}{12}$, the magnitude of the covariance dominates the width of the effective range. Therefore, the larger covariances observed in the InceptionV3-ImageNet setting also explains the broader effective region demonstrated in Figure 6b.

### D.4. Additional Qualitative Examples

Due to space constraints, additional sample explanations could not be included in Section 5. To better illustrate the explanations derived from the proposed framework and visually compare them to the best-performing competitors, we provide supplementary qualitative examples in the following pages. These sample explanations cover all tested cases and are organized according to the corresponding test setting. Please note the different color schemes used in the text and image examples. In text examples, blue and red indicate contributions to positive and negative answers for better comprehensibility in binary classification tasks. For images, which set up a multi-class classification problem, blue pixels represent supportive evidence for the target prediction, whereas red pixels oppose it.

Figure 8: Sample explanations derived from **BERT-Amazon**. Each block presents explanations for an entry derived by *IG*, *PSHAP*, and *GEFA*, respectively.

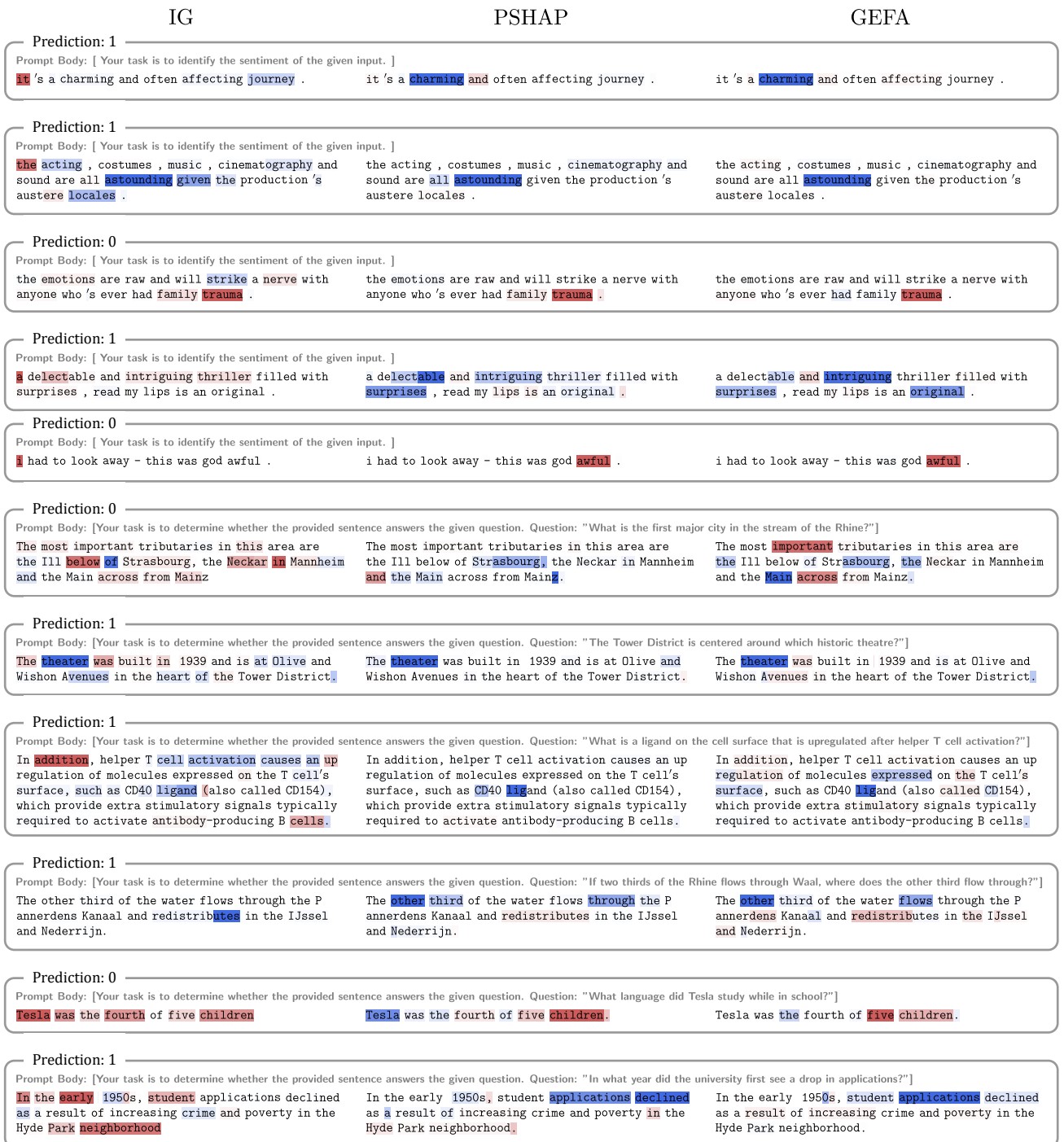

Figure 9: Sample explanations derived from **Llama3.2**. Each block presents explanations for an entry derived by *IG*, *PSHAP*, and *GEFA*, respectively. The first five examples are instances from *SST-2* dataset, while the following six originate from *QNLI*. A shortened version of the template prompt is shown in gray text. Please note that the QNLI examples differ slighly in template prompts, as the question may vary.

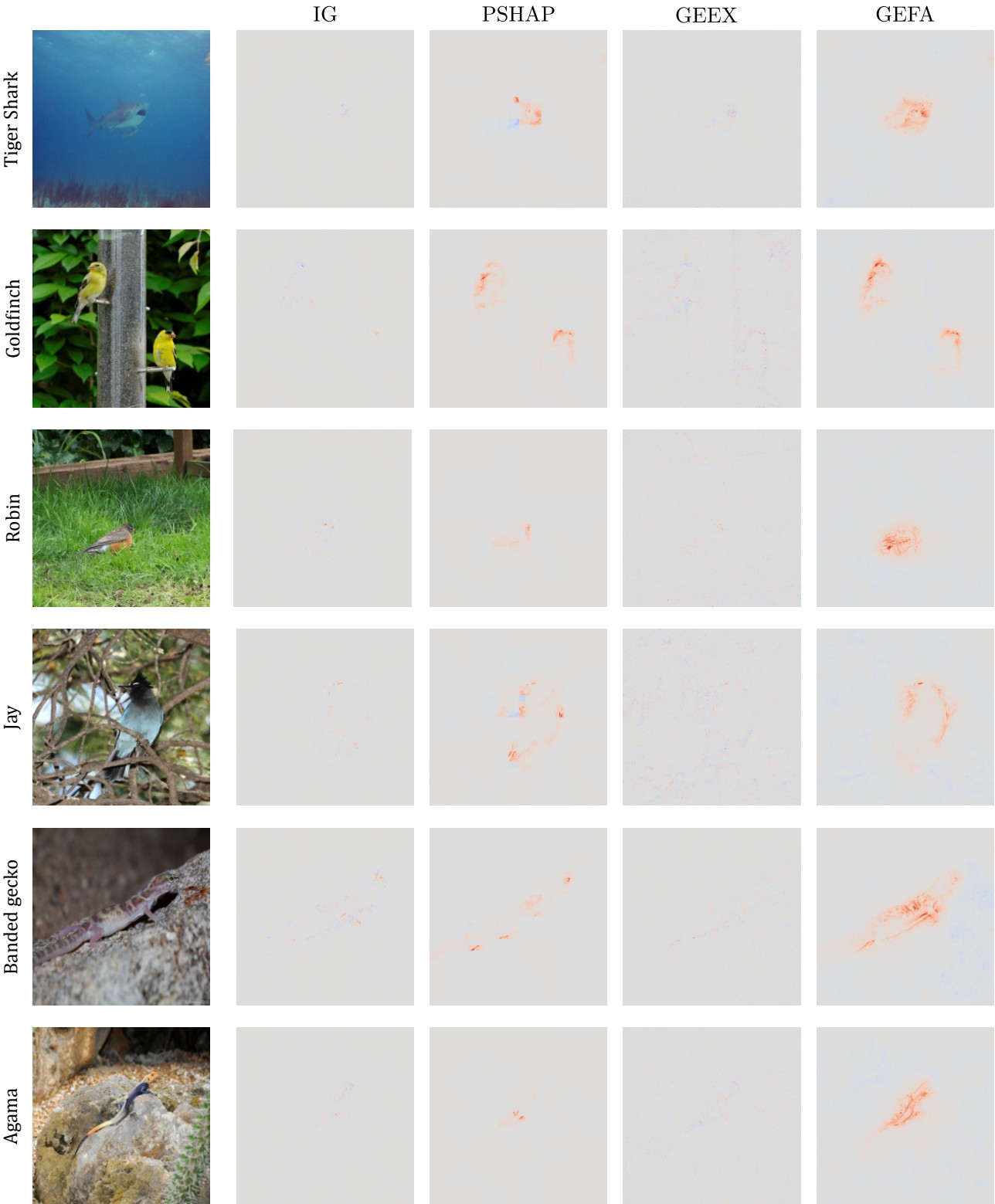

Figure 10: Sample explanations derived from **InceptionV3-ImageNet**. With explicands placed in the first column, each row presents explanations for an entry derived by *IG*, *PSHAP*, *GEEX*, and *GEFA*, respectively. The target class is shown by the class name to the left of the corresponding explicand.

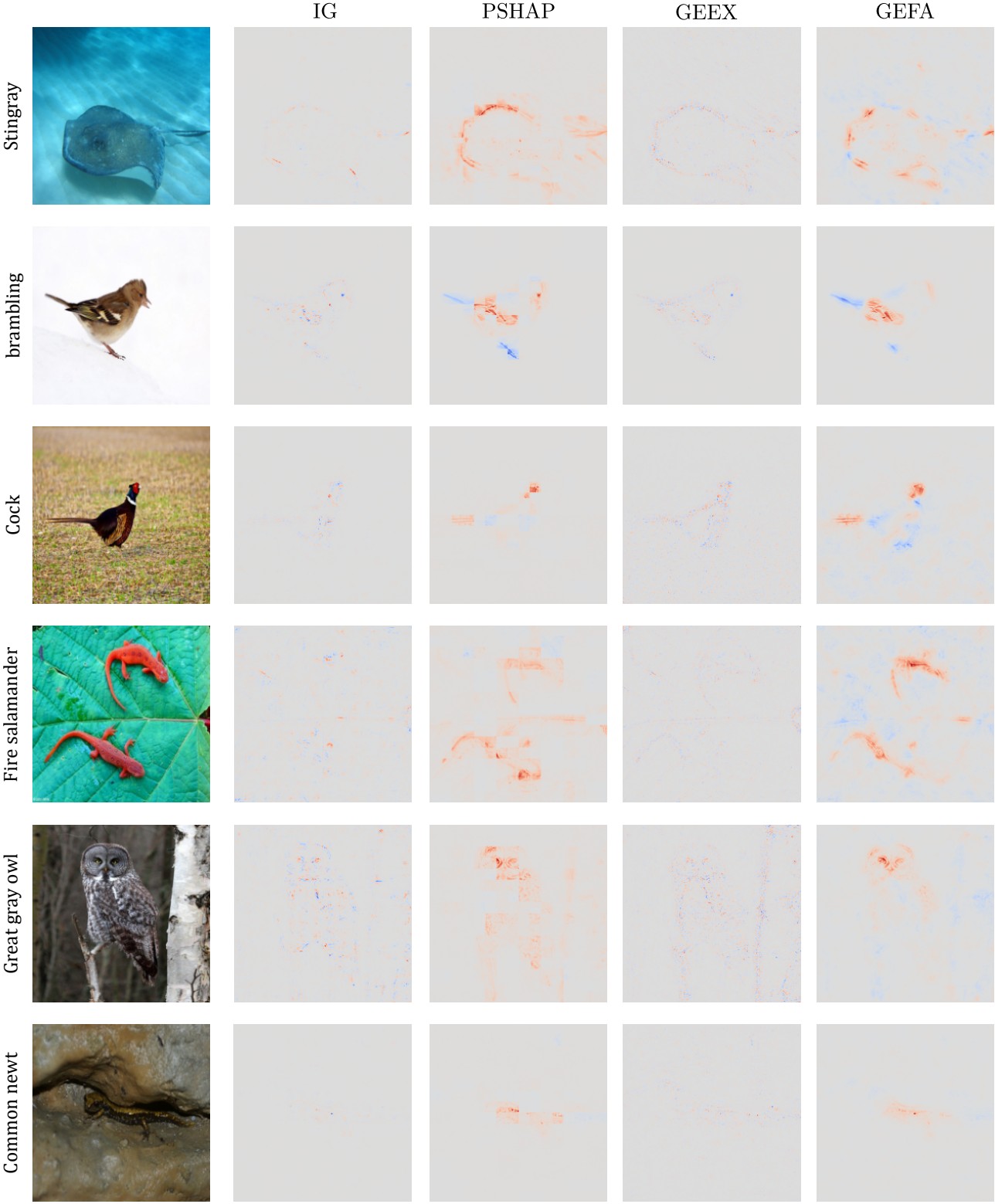

Figure 11: Sample explanations derived from **ViT-ImageNet**. With explicands placed in the first column, each row presents explanations for an entry derived by *IG*, *PSHAP*, *GEEX*, and *GEFA*, respectively. The target class is shown by the class name to the left of the corresponding explicand.

