# OpenReview forum: "GEFA: A General Feature Attribution Framework Using Proxy Gradient Estimation"
_ICML.cc/2025/Conference — ICML 2025 poster_

### Official Review · Reviewer_q8np · 2025-02-24

**Overall Recommendation:** 4

**Summary:**

This paper introduces GEFA, a feature attribution framework leveraging proxy gradients to generate explanations for different kinds of ML models. Unlike prior gradient-based explainers that operate under white-box assumptions, GEFA is designed to work for black-box models, and it is applicable to models with only query access. The method builds upon a proxy space representation, which enables estimation of feature attributions through a path integral approach, aligning with integrated gradients and providing an unbiased estimation of the Shapley value. Experiments on text (Amazon Reviews, SST-2, QNLI) and image (ImageNet) classification tasks demonstrates the effectiveness of GEFA over baselines, including IG, KernelSHAP, and GEEX.

## update after rebuttal
The response has cleared my concerns. I have no more questions and will keep my score.

**Claims And Evidence:**

Yes. The paper makes the following theoretical claims and proved them.

- GEFA is an unbiased estimator of the Shapley Value: This is mathematically demonstrated in Thm 2 and App. A1.

- GEFA is equivalent to IG when taking the same edge path: This is mathematically demonstrated in Thm 5. This connection is important for grounding GEFA in existing XAI literature.

**Essential References Not Discussed:**

The coverage of related work is reasonable.

**Experimental Designs Or Analyses:**

Yes, I checked the experiment results. They make sense. However, it's hard to say that the qualitative evaluation in Figure 2 is convincing, as these kind of visualizations were known to be misleading [1].

[1] Adebayo, J., Gilmer, J., Muelly, M., Goodfellow, I., Hardt, M., & Kim, B. (2018). Sanity checks for saliency maps. Advances in neural information processing systems, 31.

**Methods And Evaluation Criteria:**

Yes. Both text and image datasets are considered. The evaluation strategy is reasonable, using feature deletion to quantify their importance quality. The authors acknowledge potential limitations of deletion-based evaluation, e.g. concerns of OOD.

However, as an explanation work that is supposed to be deployed for humans to use, having human evaluations would strengthen the papers quality.

**Other Comments Or Suggestions:**

No more.

**Other Strengths And Weaknesses:**

Strengths:

- Solid theoretical results, with proofs supporting key claims.

- Empirical evaluations covering both text and image data.


Weaknesses:

- As an explanation work that is supposed to be deployed for humans to use, having human evaluations would strengthen the paper's quality.

**Questions For Authors:**

See above.

**Relation To Broader Scientific Literature:**

The paper is well-situated within the feature attribution and explainability literature. It builds on prior work in Shapley-based explanations (e.g. SHAP) and gradient-based attribution methods (e.g. IG). The work also contributes to the debate on black-box vs. white-box explainers, demonstrating that black-box methods can achieve competitive performance.

**Theoretical Claims:**

I scanned through the proof of Thm 2 in App. A1. Seems to be correct.

---

> ### Author Rebuttal · Authors · 2025-03-31
>
> We would like to thank the reviewers for the detailed comments and the efforts devoted to reviewing the paper. We are encouraged that our efforts in providing theoretical grounding for the proposed approach were well perceived. Our point-to-point responses to the concerns and questions raised in the comments are given below.
>
> **Sanity checks for feature attribution**: We understand the reviewer’s concern regarding the performance of the proposed approach under sanity checks. Theoretically, GEFA carries fewer risks of reproducing input information — especially when compared to white-box approaches, which can heavily rely on model structure and input values.
>
> The explanation process described by Eq. (8) delivers feature attributions according to observations of model outcomes. As such, concrete attribution scores naturally change when the learnable model parameters are altered and the outputs differ. Specifically, the computation in Eq. (8) depends on model predictions and randomly sampled masks. The resulting explanations do not explicitly rely on input values, thereby minimizing the risk of reproducing input information and reinforcing GEFA’s focus on unraveling model behaviors.
>
> To further support our claim, we performed sanity checks on the competitors considered in our work. The following table presents **Spearman Rank Correlations** between explanations derived from a pretrained model and those from a randomly initialized version of the same model architecture.
> We report the rank correlation between the absolute attribution scores determined on the two model versions. A lower correlation magnitude indicates better performance under the sanity check.
>
> |Rank Correlations|&nbsp;&nbsp;&nbsp;&nbsp;VG|&nbsp;&nbsp;&nbsp;&nbsp;IG|PSHAP|GEEX|GEFA|
> |-|-|-|-|-|-|
> |&nbsp;&nbsp;&nbsp;&nbsp;&nbsp;&nbsp;InceptionV3|0.2371|0.5701|0.4035|0.5695|**0.1249**|
> |&nbsp;&nbsp;&nbsp;&nbsp;&nbsp;&nbsp;&nbsp;&nbsp;&nbsp;&nbsp;&nbsp;&nbsp;&nbsp;ViT|**-0.0021**|0.4351|0.1253|0.5633|0.0625|
>
> Consistent with the above argument, GEFA achieves competitive performance in the sanity checks, exhibiting low correlation values in both test settings (InceptionV3 and ViT). In contrast, IG and GEEX, which explicitly incorporate input information during their explanation processes, perform relatively worse in the tests.
>
> An additional observation is that all competitors tend to obtain lower correlation scores on ViT compared to the InceptionV3 setting.
> Although this goes beyond the scope of feature attribution evaluation, we interpret this difference as a consequence of architectural characteristics inherent to CNNs. InceptionV3 depends heavily on convolution operations, which implicitly encode the prior knowledge about the relevance of spatially adjacent pixels. While this architectural bias facilitates model training and improves prediction performance, it can potentially lead to more consistent attribution patterns, even across different model versions, thereby resulting in higher explanation similarity.
> By contrast, the attention mechanism in ViT allows interactions between arbitrary features, regardless of their spatial distance. As a result, the classification behavior of ViT is more dependent on its learnable parameters rather than architectural priors. This leads to a more significant change in model behavior after random initialization, which in turn results in lower rank correlations in the sanity checks.
>
> **Human evaluation**: We thank the reviewer for the constructive comment and fully agree that explanation comprehensibility to humans represents a crucial aspect of explanation quality. However, we did not include human evaluation at the current stage of our work due to concerns about human inductive bias. Without specific knowledge of the tested model, human evaluators may form expectations that diverge from the underlying model behaviors. For example, human evaluators may expect feature attributions to highlight the target object, potentially underestimating the quality of explanations derived for models that rely on features different from those anticipated by humans.
>
> With the effectiveness of GEFA demonstrated through automated evaluations, we will carefully consider incorporating human evaluations in future work to improve the presentation of explanation results. We also aim to further explore potential utilities of feature-attribution-based explanations in understanding and improving data-driven models, particularly in the aspects of debugging and debiasing.
>
> We hope that our responses address the concerns that the reviewer has raised. We look forward to further comments from the reviewer and are ready to engage in the next round of discussion.

---

> > ### Comment · Reviewer_q8np · 2025-04-02
> >
> > Thanks! The response has cleared my concerns. I have no more questions and will keep my score.

---

### Official Review · Reviewer_fwuD · 2025-03-09

**Overall Recommendation:** 3

**Summary:**

This work presents GEFA  -- Gradient-estimation-based Explanation For All. GEFA is a general feature attribution framework based on proxy gradient estimation. The authors argue that GEFA offers a black-box explainability solution that is broadly applicable across different input modalities (e.g., images, text) while maintaining theoretical guarantees. The main contributions are (1) A new black-box feature attribution method leveraging proxy gradients. (2) A proof that GEFA produces unbiased estimates of Shapley Values. (3) A comparison between GEFA and Integrated Gradients (IG), demonstrating that the two methods coincide under specific path choices. (4) Empirical validation intended to show improved efficiency and faithfulness over existing methods.

**Claims And Evidence:**

The paper’s core claim is that GEFA generalizes feature attribution beyond previous black-box methods while maintaining Shapley-based guarantees. The theoretical analysis supports this, but empirical validation has limitations, see weakness section.

**Essential References Not Discussed:**

see my previous point.

**Experimental Designs Or Analyses:**

The experiments are well-organized, althgouht they could be expanded (see my remarks below)

**Methods And Evaluation Criteria:**

The method is evaluated via Deletion based metrics on ImageNet using a InceptionV3 and a ViT. For text classification they using BERT and LLaMA.
- Single evaluation metric AOPC. I am not fully convinced by ROAR (I think it explain the distribution of functional rather than a single model) but i would like to see at least MuFidelity / Insertion and Deletion, not just normalized AOPC.
- Limited Baselines: Many black-box methods are missing. I would like to see how RISE (petsiuk), HSIC (novello) compare to your method, many method are not considered, despite their relevance.

**Other Comments Or Suggestions:**

See the previous section.

**Other Strengths And Weaknesses:**

Strengths:
- The proofs of Shapley Value equivalence and variance reduction are strong and interesting.
- Clear writing and structure, the explanations are mathematically rigorous and easy to follow.
- I really liked the Variance reduction part !

However, despite being interesting and well-written, Here are, in my opinion, the weak points of the paper, which I will group into major problems (**M**) and minor problems (**m**).

Major (**M**):

**M1**: Lack of comparisons with black-box method: lime, rise hsic. These are relevant for black-box attribution.

**M2**: Single evaluation metric. Deletion-based AOPC is common but insufficient. I would like to see Insertion, Deletion and Mufidleity.

now for the Minor (**m**):

**m1**: No failure cases. What happens when proxy gradient estimation fails?

**m2**: Discussion of hyperparameters is missing. How does query budget impact performance?

**m3**: Novel Insights into Model Behavior: A key question that i like to ask on any interpretability research is: **What new insights about model behavior does this method uncover?** If the method can reveal previously unknown biases or learned shortcuts or anything else new.

**Questions For Authors:**

See **M1,2** and **m1,2,3**.

**Relation To Broader Scientific Literature:**

Medium, 34 citations are good, but I think we could ask for more given the number of articles in this area, and for ICML. Especially, many black-box attribution methods and metrics are missing: Bhatt et al., 2020; Jacovi & Goldberg, 2020; Hedstrom et al. 2022; Hsieh et al., 2021; Boopathy et al.,2020; Lin et al., 2019;  Fel et al., 2021. Idrissi et al., 2021; Novello et al., 2022.

**Theoretical Claims:**

The theoretical contributions are solid.

---

> ### Author Rebuttal · Authors · 2025-03-31
>
> We would like to thank the reviewers for the detailed comments and the efforts devoted to reviewing the paper. We are encouraged that our theoretical grounding for the proposed approach was well received and that the reviewer liked the analyses. Our point-to-point responses to the concerns and questions are given below.
>
> First, we appreciate the thoughtful reference list, which will help improve the related work section with more thorough coverage of SOTA.
>
> **M1**: In the experimental section, we initially focused on gradient-based methods and the SHAP family, given their respective connections to gradient estimation and Shapley Values. However, we agree that RISE and HSIC are representative and relevant competitors in the black box setting, particularly due to their use of binary masks for query generation. We have expanded our experiments to include them.
>
> **M2**: We have extended the experiments with Insertion, Deletion, and MuFidelity. For the two additional competitors, we use the author-released version of RISE and ***xplique*** for HSIC.
> Please note that we use AUC to quantify method performance in Insertion and Deletion, strictly following the setting used in HSIC (novello). Lower values are better for Deletion (indicated by ↓), whereas higher values are better for Insertion and μFidelity (indicated by ↑).
>
> For a better sense of the results, we first evaluated the competitors on ResNet50 with a zero baseline — a setting used by RISE and HSIC. GEFA performs competitively across all tests, consistently ranking among the top two (in bold).
> |ResNet50|&nbsp;&nbsp;&nbsp;&nbsp;IG|PSHAP|GEEX|GEFA|RISE|HSIC|
> |-|-|-|-|-|-|-|
> |Deletion ↓|**0.0493**|0.1691|0.0921|**0.0750**|0.1073|0.0890|
> |Insertion↑|0.1888|0.3030|0.2844|**0.7297**|**0.6395**|0.5831|
> |μFidelity ↑|0.0328|0.0259|0.0241|**0.0629**|0.0428|**0.0673**|
>
> The tests were also repeated on InceptionV3 with the original setting from our paper. The results align with those reported using nAOPC scores. This is expected, as nAOPC and AUC (used by Deletion and Insertion) measure complementary areas along the perturbation curve.
> |Inception|&nbsp;&nbsp;&nbsp;&nbsp;IG|PSHAP|GEEX|GEFA|RISE|HSIC|
> |-|-|-|-|-|-|-|
> |Deletion ↓|**0.0926**|0.1803|0.1661|**0.1046**|0.2205|0.1674|
> |Insertion↑|**0.8048**|0.7532|0.7421|**0.8235**|0.7244|0.6926|
> |μFidelity ↑|**0.0851**|0.0407|0.0441|**0.0519**|0.0136|0.0401|
>
> Due to space constraints, we cannot provide a full view of our understanding of the experimental results. However, we welcome any further questions and would like to explore additional insights together.
>
> **m1 (Failure case)**: Proxy gradient estimation can face challenges when feature masking has minimal impact on model outputs. Such cases may arise when the classification target is represented by redundant or widely distributed features — masking only parts of relevant features fails to expose model sensitivity, despite their relevance. This can lead to an underestimation of attributions to truly relevant features, resulting in noisy and less comprehensible explanations. The most straightforward solution is to enlarge the query budget, which increases the chance of sampling effective masks that expose model sensitivities. In addition, the use of mask smoothing (Appendix B.3.3) mitigates the risk of such failure cases. By softly grouping locally adjacent features, mask smoothing increases the probability of removing meaningful local patterns, inducing more significant changes in model outcomes.
>
> **m2 (Hyperparameters)**: The query budgets are empirically determined based on the feature space dimensionality of each test case. Appendix D.1 investigates and discusses GEFA’s sensitivity to the query budget. Appendices D.2 and D.3 further examine the effect of the control variate coefficient, demonstrating the optimality of the estimated $\beta^*$ and highlighting the importance of the correlation assumption stated in Assumption 3.
>
> **m3 (New insights)**: We focus on improving the quality of black-box explanations. With explanations that better reflect attributions to specific features, more faithful insights to model behaviors become available, thereby contributing to specific use cases, e.g. debugging and debiasing noted by the reviewer, and potentially model distillation (more effective model knowledge transferring by encouraging focus on salient regions). Additionally, we see GEFA’s potential to handle more complicated outcomes (e.g. text generation by LLMs). This is inspired by the use of gradient estimation in managing delayed rewards in RL — a challenge where backpropagation is less effective. We are currently investigating better formulations for model outcome observations and exploring the compatibility of GEFA with models producing more complex outputs, including those involving multi-modalities.
>
> We hope our responses adequately address the reviewer's concerns. We look forward to further comments and are ready to engage in the next round of discussion.

---

> > ### Comment · Reviewer_fwuD · 2025-04-04
> >
> > Thank you for the thorough and thoughtful response.
> > I appreciate the added experiments and clarifications—you've addressed my concerns well.
> >
> > The method is sound and the results are clearer now. My only remaining hesitation is the relatively limited impact (attribution methods), which is why i’m not increasing my score further. Again, congratulations to the authors, I wish them best of luck with the paper acceptance !

---

### Official Review · Reviewer_xeh5 · 2025-03-14

**Overall Recommendation:** 4

**Summary:**

In this work, the authors propose a blackbox feature attribution method based on proxy gradient estimation. Specifically, they introduce proxy variables, each denoting a binary feature-level selection. The authors show that their approach is an unbiased estimator of shapley values, thus sharing some of the nice properties of shapley values. They also show experimentally that their method works similarly to integrated gradient method, where gradients can directly be estimated.

**Claims And Evidence:**

Claims are supported by theoretical results and empirical evidence

**Essential References Not Discussed:**

- Literature on issues with feature attribution based explanations e.g. [1] not linked to very well

Adebayo, Julius, et al. "Sanity checks for saliency maps." Advances in neural information processing systems 31 (2018).

**Experimental Designs Or Analyses:**

- The work utilizes standard datasets used for such attribution-level expeiments (SST2).While they compare different blackbox models for text, they only consider Inceptionv3 for images. The setup makes sense, though the datasets/models used are relatively small. However, they show the utility of the method
- Can authors discuss more about connections to linear regression (e.g. similar to SHAP), and some local explanations like LIME -- which also uses the idea of masking variables but is not gradient-based/has guarantees

**Methods And Evaluation Criteria:**

- The work utilizes standard datasets used for such attribution-level expeiments (SST2).While they compare different blackbox models for text, they only consider Inceptionv3 for images.
- Generally experimental methodology makes sense, however quantitative results focus on a single metric. More evaluation metrics (e.g. impact of data perturbations to accuracy), similar to prior work (Do Feature Attribution Methods Correctly Attribute Features?, Zhou et al, AAAI 2022) would add to the experimental evidence

**Other Comments Or Suggestions:**

- Can authors expand on limitations of this work/using feature attributions as interpretations?

**Other Strengths And Weaknesses:**

Experimental methodology seems sound, though results are primarily on relatively small datasets. Can authors comment more on the computational complexity?

**Questions For Authors:**

- Can authors discuss more about connections to linear regression (e.g. similar to SHAP), and some local explanations like LIME -- which also uses the idea of masking variables but is not gradient-based/has guarantees
- Can authors comment on feasibility with larger datasets?

**Relation To Broader Scientific Literature:**

- The paper proposes a proxy-gradient estimation based method for feature attribution, and connect it to prior work in the feature attribution based explanation space

**Theoretical Claims:**

Checked A.1 (equivalence to shapley values), and it seems correct (but I've not thoroughly verified in detail)

---

> ### Author Rebuttal · Authors · 2025-04-01
>
> We would like to thank the reviewers for the detailed comments and the efforts devoted to reviewing the paper. Our point-to-point responses to the concerns and questions are given below.
>
> **Test setting for images**: We consider two image classifiers — InceptionV3 and ViT — as shown in Table 2. These models incorporate two widely used architectural components: convolutional layers and attention mechanisms. We believe the structural diversity better demonstrates GEFA’s independence from specific model architectures.
>
> **More evaluations**: We evaluate the impact of explanation-guided manipulation on model accuracy and present the results in the following table. Prediction accuracy is reported after removing the top 50% of the most important features as identified by each explainer; lower is better.
> The impacts on prediction accuracies generally align with the nAOPC scores, which summarize the overall perturbation process.
> It is noteworthy that GEFA outperforms IG in the test as it has more comprehensive coverage of relevant features, which is shown by the perturbation curves in our response to Reviewer **yynM** (via anonymous link).
> We also refer the reviewer to our response to Reviewer **fwuD** for further results under extended settings.
> |InceptionV3|VG|IG|PSHAP|GEEX|GEFA|
> |-|-|-|-|-|-|
> |Accuracy|0.494|0.142|0.195|0.163|**0.092**|
>
> **Connections to linear regression**: Linear regression is used as a surrogate to approximate local model behaviors. While GEFA does not build a surrogate model, an indirect connection is drawn by KernelSHAP, which shows that linear regression with a carefully designed weighting kernel can serve as an estimator of Shapley Values.
>
> **Connections to local explanations**: As noted by the reviewer, GEFA shares a high-level idea with other black-box methods. However, GEFA distinguishes itself from heuristic-based approaches through the rigorously derived sampling strategy and observation aggregation process. These analyses further provide theoretical grounding and desirable properties for the proposed method.
>
> **Sanity checks**: We understand the reviewer’s concern about sanity checks. Theoretically, GEFA derives explanations based on observations of model outputs; thus, concrete attribution scores will change when model parameters are altered, as it results in different predictions. Due to character limitations in the response, we kindly refer the reviewer to our reply to Reviewer **q8np** for further details and experimental results.
>
> **Time complexity**: Appendix B.3.4 provides the time complexity analysis from the perspective of query budgets. Generally, black-box competitors exhibit similar complexity when receiving identical query budgets. However, some methods involve additional steps during explanation process, which can be slower than GEFA in practice. From the perspective of feature space dimensionality, the query search space grows exponentially as the feature space expands, posing a challenge for all black-box explainers. We incorporate *mask smoothing* (Appendix B.3.3) to counteract the complexity due to the increase of feature space dimensionality.
>
> **Feasibility with larger datasets**: We interpret “larger datasets” in two ways: datasets with more entries and inputs with more features. GEFA is insensitive to dataset size, as feature attribution focuses on explaining individual decisions. In contrast, larger inputs indeed increase computational complexity for black-box explainers, consistent with the discussion about time complexity.
>
> **Limitations**: Feature attribution is an important step toward understanding model behavior. However, further developments are awaited for deeper insights. Current approaches typically investigate ultimate feature contributions to model outcomes without considering interactions among features. This can conceal details about how models interpret inputs. For example, CNNs and transformers process inputs differently, but such differences are often not perceptible from feature attribution alone. We are looking into the potential of taking higher-order derivatives within the GEFA framework to reveal interactions between active features. Additionally, the rise of LLMs presents new challenges for explainability. While we obtained promising results with simple test cases on LLMs, we believe that feature attribution is only one piece of the explainability puzzle. Unlike classifiers that typically receive inputs with sufficient information for prediction, LLM prompts often pose questions that require the model to draw on knowledge acquired during training. We believe that feature attribution should at least be complemented by data attribution to demonstrate: 1) how a model interprets a given prompt; 2) which parts of training data contribute to the knowledge for model reactions.
>
> We hope our responses adequately address the reviewer’s concerns. We look forward to further feedback from the reviewer and are ready to engage in continued discussion.

---

### Official Review · Reviewer_yynM · 2025-03-15

**Overall Recommendation:** 3

**Summary:**

In this paper, the authors propose a new method for input attribution in DNNs. They focus on the attribution in black-box models, where the gradient is unavailable. In this case, they propose the proxy gradient space for estimation, and then define the attribution of input features. The authors further prove the properties of the proposed metric. They also modify the metric for further variance reduction.

**Claims And Evidence:**

No.

**Essential References Not Discussed:**

No.

**Experimental Designs Or Analyses:**

- How many queries were used for GEFA and other baseline methods in experiments? Is GEFA the best under the same number of queries?

**Methods And Evaluation Criteria:**

- The advantage of GEFA over previous methods is not significant, especially SHAP. From the perspective of correctness, GEFA is an unbiased estimation of SHAP. From the perspective of effectiveness, the theoretical complexity of Eq. (7) is larger than SHAP, since Eq. (7) additionally involves the integration over $\gamma$. Besides, the “information waste” is claimed as a shortcoming of SHAP but I am not sure what it exactly means.

- In the integrating path of GEFA, all input features share the same presence probability, $\forall i, \alpha_i = \gamma$. What is the benefit of such a setting and why not use different presence probabilities?

- The evaluation of attribution methods is limited to nAOPC. Evaluation based on insertion and sanity checks in (Adebayo et al., 2018) should be included. Moreover, besides the nAOPC values, the change curve of model performance along with the deletion should be reported.

Adebayo et al., Sanity checks for saliency maps. In NeurIPS 2018.

**Other Comments Or Suggestions:**

The comparison of computation cost of different methods is suggested to be put in the main text.

**Other Strengths And Weaknesses:**

No.

**Questions For Authors:**

No.

**Relation To Broader Scientific Literature:**

This paper provides a potential method for estimating attributions in black-box models, extending the integrated gradient to the black setting.

**Theoretical Claims:**

The proof of the completeness in Appendix A.2.1 is confusing. Notations like $w_{i\in S}$ and $w_{i\notin S}$ are not formally defined, and equations in Lines 609-612 need explanations.

---

> ### Author Rebuttal · Authors · 2025-03-31
>
> We would like to thank the reviewers for the detailed comments and the efforts devoted to reviewing the paper. Our point-to-point responses to the concerns and questions are given below.
>
> We start with the relationship between GEFA and SHAP:
> 1. Both GEFA and SHAP are unbiased estimators of Shapley Values, rather than one being an estimator of the other. They are closely related, as both theoretically converge to the same results when the query budget increases.
> 2. Appendix B.3.4 provides analyses of **time complexity** for the black-box competitors. When the number of queries is set to $n$, the black-box competitors (including GEFA) exhibit the same level of complexity. For GEFA, the query budget is distributed across points on the proxy path, but the total number of queries sticks to $n$. KernelSHAP additionally solves a linear regression problem, and PartitionSHAP applies an explicand-specific feature space partitioning; both introduce additional costs. These costs become more significant as feature space grows, which is empirically demonstrated by Table 4 in Appendix B.3.4.
> 3. We refer to “information waste” as a limitation of Shapley value estimators relying on computing marginal contributions. Let $x_i$ be a present feature in an observation $x_S$. To compute its marginal contribution in the context of $S$, a paired observation $x_{S\backslash i}$, differing exactly in $x_i$, is required. In other words, for any present feature $x_j$, its marginal contribution given $S$ cannot be computed if model prediction on $x_{S\backslash j}$ is not observed. This induces information waste, as the information about $x_j$ contained in $x_S$ is not used. Regardless of sampling strategies, such waste is unavoidable unless all possible combinations of feature presence are enumerated, which becomes computationally intractable in high-dimensional feature spaces.
> 4. Furthermore, we would like to highlight the explicit gradient-estimator form of GEFA, which enables the application of the **control variate**. The designed control variate further reduces the estimation variance without additional queries, thereby improving explanation quality. This advantage is theoretically proved and empirically shown in our experiments and Appendix D.2.
>
> **Proxy path**: As discussed in the last paragraph of Section 4.4, the straight-line path — where all $\alpha_i$ takes the same value — is equivalent to averaging over all $p!$ unique edge paths. The equivalence allows us to simplify the problem of averaging estimates across multiple paths to computing a single path-based estimate.  For further details, we refer the reviewer to Section 4.4 and Appendix A.4.
>
> **Further evaluations**: We have conducted the additional tests suggested by the reviewer. Due to character limitations in this response, we kindly invite the reviewer to refer to our reply to Reviewer **q8np** for details on sanity checks, and our reply to Reviewer **fwuD** for the Insertion test and other expansions.
>
> **Change curves**: Change curves provide additional insights into explanation effectiveness. We will include plots showing the changing trends under different test settings in the Appendix of the updated version. A preliminary version of change curves is available via https://anonymous.4open.science/api/repo/change_curves_GEFA-F417/file/curves.pdf?v=a9189c14
>
> **Notion in Appendix A.2.1**: We use $w_{i\in S}$ and $w_{i\notin S}$ as shorthand for the weighted contributions of observation to the feature attribution estimate of $x_i$, as defined in the equations in lines 609-612. When combined, the two parts recover Eq. (8). However, we noticed that the definitions mistakenly use $=$ instead of the assignment symbol $:=$. We believe this notation mistake likely caused the confusion, and we thank the reviewer for pointing it out. The notation will be corrected in the updated version.
>
> **Number of queries**: All black box competitors receive identical query budgets. GEFA outperforms other black-box explainers given the same query budgets. Specifically, the number of queries is 500 for text classifiers (lines 290-291, right panel) and 5000 for image classifiers due to the higher dimensionality of images (lines 369-370, left panel). Additionally, Appendix D.1 provides further results evaluating GEFA under varying query budgets. In light of the reviewer’s comment, we recognize that the query budgets should be better highlighted. We will consolidate and relocate the information to the experimental setting section.
>
> **Computation cost**: We agree that computational cost is an important aspect in comparing black-box explainers. While the current version presents this information in the Appendix due to space constraints, we appreciate the suggestion and will move the relevant results to the main text if space permits.
>
> We hope our responses adequately address the reviewer's concerns. We look forward to further feedback and are ready to engage in continued discussion.

---

### Decision · Program_Chairs · 2025-05-01

**Decision:**

Accept (poster)

**Comment:**

This paper introduces GEFA, a black-box feature attribution framework based on proxy gradient estimation that provides unbiased Shapley value estimates. The approach connects closely with Integrated Gradients while avoiding dependence on direct model gradients, thereby extending applicability to black-box settings. The paper is theoretically sound, and the authors provide proofs for their core claims.
The reviewers appreciated the theoretical contributions, clean writing, and extensive empirical evaluation. While early reviews raised concerns about limited evaluation metrics and missing comparisons to other black-box explainers (e.g., RISE, HSIC), the authors addressed these issues thoroughly during the rebuttal by adding new experiments (e.g., insertion, deletion, MuFidelity) and additional baselines. Reviewers also asked for clarifications regarding complexity, proxy paths, and practical limitations, which were addressed in a detailed rebuttal. While some reviewers noted that the overall impact could be incremental, all acknowledged the method as technically sound, well-executed, and likely to be a useful tool in the community.